

# Model evaluation and inter-comparison of surface-level ozone and relevant species in East Asia in the context of MICS-Asia phase III Part I: overview

Jie Li[1,2,3], Tatsuya Nagashima[4], Lei Kong[1,2], Baozhu Ge[1,2,3], Kazuyo Yamaji[5], Joshua S. Fu[6], Xuemei Wang[7], Qi Fan[8], Syuichi Itahashi[9], Hyo-Jung Lee[10], Cheol-Hee Kim[10], Chuan-Yao Lin[11], Meigen Zhang[1,2,3], Zhining Tao[12], Mizuo Kajino[13,14], Hong Liao[15], Meng Li[16], Jung-Hun Woo[10], Jun-ichi Kurokawa[17], Qizhong Wu[18], Hajime Akimoto[4], Gregory R. Carmichael[19] and Zifa Wang[1,2,3]

[1]LAPC, Institute of Atmospheric Physics, Chinese Academy of Sciences, Beijing, 100029, China

[2]College of Earth Sciences, University of Chinese Academy of Sciences, Beijing, 100049, China

[3]Center for Excellence in Urban Atmospheric Environment, Institute of Urban Environment, Chinese Academy of Sciences, Xiamen, 361021, China

[4]National Institute for Environmental Studies, Onogawa, Tsukuba, 305-8506, Japan

[5]Graduate School of Maritime Sciences, Kobe University, Kobe, 657-8501, Japan

[6]Department of Civil and Environmental Engineering, University of Tennessee, Knoxville, TN, 37996, USA

[7]Institute for Environment and Climate Research, Jinan University, Guangzhou, 510632, China

[8]School of Atmospheric Sciences, Sun Yat-sen University, Guangzhou, 510275, China

[9]Central Research Institute of Electric Power Industry, Tokyo, 100-8126, Japan

[10]Department of Atmospheric Sciences, Pusan National University, Pusan, 46241, South Korea

[11]Research Center for Environmental Changes/Academia Sinica, 11529, Taipei

[12]Universities Space Research Association, Columbia, MD, 21046, USA

[13]Meteorological Research Institute, Tsukuba,305-8506, Japan

[14]Faculty of Life and Environmental Sciences, University of Tsukuba, Tsukuba, 305-8506, Japan

[15]Jiangsu Key Laboratory of Atmospheric Environment Monitoring and Pollution Control, Jiangsu Collaborative Innovation Center of Atmospheric Environment and Equipment Technology, School of Environmental Science and Engineering, Nanjing University of Information Science & Technology, Nanjing, 210044, China

[16]Ministry of Education Key Laboratory for Earth System Modeling, Department of Earth System Science, Tsinghua University, Beijing, 100084, China

[17]Japan Environmental Sanitation Center, Asia Center for Air Pollution Research, Niigata, 950-2144, Japan

[18]Beijing Normal University, Beijing, 100875, China

[19]Center for Global and Regional Environmental Research, University of Iowa, Iowa City, IA, 52242, USA

*Correspondence to*: Jie Li (lijie8074@mail.iap.ac.cn)



**Abstract**: Long-term ozone ($O_3$) and nitrogen oxide ($NO_x$) from fourteen state-of-the-art chemical

transport models (CTMs) are evaluated and intercompared to $O_3$ observations in East Asia, within the

framework of the Model Inter−Comparison Study for Asia phase III (MICS-ASIA III), designed to

evaluate the capabilities and uncertainties of current CTMs simulations in Asia and provide multi-model

estimates of pollutant distributions. These models were run by fourteen independent groups working in

China, Japan, South Korea, the United States and other countries/regions. Compared with MICS-Asia II,

the evaluation against observations was extended to be one-full year in China and the western Pacific

Rim from four months and the western Pacific Rim. Potential causes of discrepancies between model

and observation were investigated by assessing the planetary boundary layer heights, emission fluxes,

dry deposition, $O_3$-$NO_x$ relationships and vertical profiles as determined by the models in this study. In

general, the model performance levels for $O_3$ varied widely, depending on region and seasons. Most

models captured the key pattern of monthly and diurnal variation of surface $O_3$ and its precursors in North

China Plain, Yangtze River Delta and western Pacific Rim, but failed in Pearl River Delta. A significant

overestimation of surface $O_3$ was evident in May-September/October and January-May over the North

China Plain, western Pacific Rim and Pearl River Delta. A large intermodel variability of $O_3$ existed in

all subregions over East Asia in this study, which was caused by the internal parameterizations of

chemistry, dry deposition and vertical mixing of models, even though the native schemes in the models

were similar. The ensemble average of 13 models for $O_3$ did not always exhibit a superior performance

compared to certain individual model, in contrast to its superiority in Europe. This suggested that the

spread of ensemble-model values had not represented all uncertainties of $O_3$ or most models in MICS-

ASIA III missed key processes. Compared with the previous phase of MICS-Asia(MICS-Asia II), this

study improved the performance of modeling $O_3$ in March at Japanese sites. However, it overpredicted

surface $O_3$ concentrations in western Japan in July, which has not been found in MICS-Asia II. Major

challenges still remain in regard to identifying the sources of bias in surface $O_3$ over East Asia in CTMs.



## 1. Introduction:

Tropospheric ozone ($O_3$) is a significant secondary air pollutant produced through thousands of photochemical reactions and detrimental to human health, ecosystems, and climate change as a strong oxidant (WHO, 2005; The Royal Society, 2008). With the fast industrialization and urbanization in the last two decades, $O_3$ concentration is rising at a higher rate in East Asia than other regions and 30% of the days in megacities (e.g. Beijing, Shanghai Guangzhou in China) exceeds air quality standard of World Health Organization (Wang et al.,2017) for 8-hour average surface $O_3$ concentration. The high $O_3$ concentrations received more attention from the public and from policy-makers in East Asia. The Ministry of Environment Japan has imposed stringent measures to reduce traffic emissions since 1990s, and non-methane volatile organic compounds (NMVOCs) and $NO_x$ mixing ratios have decreased by 40-50 % and 51-54 %, respectively (Akimoto et al.,2015). In 2012, China released a new ambient air quality standard in which the 8-hour $O_3$ maximum was set limits for the first time. However, these measures don't prevent the persistent increase of the ground-level $O_3$ in East Asia. The average mixing ratio of $O_3$ increased 20-30% in Japan over the last 20 years (Akimoto et al.,2015). In Chinese megacities, 8-hr $O_3$ concentrations have increased 10-30 % since 2013(Wang et al.,2017).

The primary method for detailed evaluation of the effect of air quality policies at the scale of East Asia is numerical air quality modeling. Several global and regional scale CTMs (e.g. GEOS-Chem, CHASER, CMAQ, CAMx, WRF-Chem and NAQPMS) over the past few decades have been developed and widely used to simulate the $O_3$ formation process and evaluate its control strategies (Streets et al., 2008; Li et al., 2007; 2008; Yamaji et al., 2006;Zhang et al., 2008;Liu et al., 2010; Wang et al., 2013; He et al., 2017; Nagashima et al.,2017). These simulations have identified the key precursors of $O_3$ formation in East Asia (Zhang et al.,2008; Liu et al., 2010; Tang et al., 2011; He et al., 2017), assessed the contributions of international and regional transport (Streets et al., 2008; Li et al., 2008), and predicted the $O_3$ mixing ratios in different future emission scenarios (Wang et al., 2013). However, discrepancies remain between models and observations, indicating that model simulations of $O_3$ in East Asia still need to be improved (Han et al., 2008). Modeling uncertainties related to the emissions, chemistry, wet and dry deposition, and transport can hardly be handled using one single model. Model inter-comparison has thus been recognized as an effective way to address problems and has been successfully applied in Europe



and North America in the phase 2 of the Air Quality Model Evaluation International Initiative (AQME II; Rao et al., 2011). Limited model inter-comparison related to air quality in East Asia has been conducted. Phases I and II of the Model Inter–Comparison Study for Asia (MICS-Asia) were initiated in 1998 and 2003, and to explore the potential sources of model uncertainties regarding sulfur, $O_3$, nitrogen

compounds and aerosols (Carmichael et al., 2002 ,2008). They found that the predicted temporal variations of surface $O_3$ in eight regional CTMs generally tended to be lower than that observed in 2001 with poor correlations in the western Pacific in March and December (Han et al., 2008). Model performance levels for $O_3$ varied largely in southern China. The inconsistency of horizontal grids, emissions and meteorological inputs among models increased the difficulty of explaining intermodel

variability in the MICS-Asia II. More importantly, model evaluation in industrialized China has not been conducted because of few observations, which has been detrimental to efforts to improve model performance levels on $O_3$.

Recently, regional CTMs have been greatly improved by coupling more mechanisms (e.g. heterogeneous chemistry and on-line calculation of photolysis rates) and accurate chemical reaction rates.

For example, the gas-phase chemistry mechanisms in Models 3-Community Multiscale Air Quality (CMAQ) have been developed into CBM05 and SAPRC07 from CB04 and SAPRC99. It is critical to evaluate the updated models' abilities for simulating current air quality over East Asia. In 2010, MICS-Asia was expanded to Phase III, in which 13 regional CTMs and 1 global CTM are run over one-full year by 14 independent groups from East Asia and North America, under a common reference model input

data set (namely, the emission inventory, meteorological fields and horizontal grids). In addition to observations made in Japan by the Acid Deposition Monitoring Network in East Asia (EANET) that were used in MICS-Asia II, new observational data from China were made available for MICS-Asia III, which were obtained from the Chinese Ecosystem Research Network (CERN) and the Pearl River Delta Regional Air Quality Monitoring Network (PRD RAQMN). An intercomparison of CTMs in China,

Japan and western Pacific over one full year has never been performed, which provided a wider database to use in the comparisons. The completeness of MICS-Asia III is therefore unique.

In this paper, we mainly evaluate the abilities of participating models in MICS-Asia III for simulating the concentration of $O_3$ and its related species in the framework of MICS-Asia III. Several



questions are addressed: (1) What is the performance level of various air quality models for simulating

$O_3$ in East Asia? (2) How consistent or discrepant are the models? (3) What are the potential factors

responsible for differences and deviations between model results and observations? (4) How do muti-

model ensembles improve the simulation accuracy for $O_3$? This paper is expected to provide valuable

insights into the abilities and limitations of CTMs in East Asia.

## 2.    Models and data

### 2.1    Experimental set up

In this study, all participating models were run for the year 2010 and provide gridded monthly mean

diurnal $O_3$ and its precursors mixing ratios in the lowest model layer. For $O_3$, monthly three-dimensional

data were also submitted. Surface concentrations were interpolated to the monitoring locations for the

model evaluation.

### 2.2    Participating models and input data

Table 1 summarizes the specifications of participating CTMs. These models include two versions

of CMAQ (v4.7.1 and 5.0.2;Byun and Schere, 2006), the Weather Research and Forecasting model

coupled   with   Chemistry   (WRF-Chem;   http:/www.acd.ucar.edu/wrf-chem),   Nested   Air   Quality

Prediction Modeling System (NAQPMS; Li et al.,2007), the Japan Meteorological Agency (JMA)'s non-

hydrostatic meteorology-chemistry model (NHM-Chem; Kajino et al., 2012), the NASA-Unified

Weather   Research   and   Forecasting   (NU-WRF;   Tao   et   al.,2013)   and   GEOS-Chem

(http://acmg.seas.harvard.edu/geos/). They have been documented in the scientific literature and widely

applied in modeling studies over East Asia. Table 1 did not list model names to maintain model

anonymity for each participating model. Similar behavior was also found in MICS-Asia II and other

model intercomparison projects (e.g. AQME II).

MICS-Asia III participants were provided with a reference meteorological field  for the year 2010,

generated with the Weather Research and Forecasting Model (WRF) version 3.4.1 model (Fig.1). WRF

v3.4.1 are driven by the final analyses dataset (ds083.2) from the National Centers for Environmental

Prediction (NCEP), with 1° × 1° resolution and a temporal resolution of 6 h. A four-dimensional data





assimilation nudging toward the NCEP dataset was performed to increase the accuracy of WRF. The horizontal model domain, which is 182 ×172 grids on a Lambert conformal map projection with 45-km horizontal resolution, is shown in Fig. 1. Vertically, the WRF grid structure consists of 40 layers from the surface to the model top (10 hPa.). The standard meteorological fields were applied by the majority

of groups. Several other models performed simulations using their own meteorological models (e.g., RAMS-CMAQ and GEOS-Chem). The WRF-Chem utilized the same model (WRF) as the standard meteorological simulation, but they considered the feedback of pollutants to the meteorological fields. Consequently, their meteorological fields are possible slightly different from the standard. GEOS-Chem is driven by the GEOS-5 assimilated meteorological fields from the Goddard Earth Observing System of

the NASA Global Modeling Assimilation Office. The couples of meteorological and CTMs vary for each group, likely resulting in a diversified set of model output.

MICS-Asia III provided a set of monthly anthropogenic emission inventory for the year 2010, which is called as MIX (Li et al.,2016). MIX is a mosaic of up-to-date regional and national emission inventories that include Regional Emission inventory in ASia (REAS) version 2.1 for the whole of Asia (Kurokawa

et al., 2013), the Multi-resolution Emission Inventory for China (MEIC) developed by Tsinghua University, a high-resolution $NH_3$ emission inventory by Peking University (Huang et al., 2012), an Indian emission inventory developed by Argonne National Laboratory (ANL-India, Lu et al., 2 011; Lu and Streets, 2012), and the official Korean emission inventory from the Clean Air Policy Support System (CAPSS; Lee et al., 2011). The biogenic emissions are taken from the Model of Emissions of Gases and

Aerosols from Nature (MEGAN). Hourly biogenic emissions were obtained for the entire year of 2010 using version 2.04 (Guenther et al., 2006). Biomass burning emissions were processed by re-gridding the Global Fire Emissions Database version 3 (GFEDv3) (0.5 by 0.5 degree). Volcano $SO_2$ emissions were provided, with a daily temporal resolution by the Asia Center for Air Pollution Research (ACAP). The emission group in MICS-ASIA III directly prepared a gridded inventory according to the configuration

of each CTM. NMVOC emissions are spectated into model-ready inputs for three chemical mechanisms: CBMZ, CB05 and SAPRC-99. Weekly and diurnal profiles were also provided. The standard emission inventory was applied by all models. The majority of models employed official suggested vertical and time profiles of pollutants from each sector by the emission group. Several other models made the



projection by themselves. More information can be found in the paper of Li et al. (2017) and Gao et al. (2017).

MICS-Asia III also provided two sets of chemical concentrations at the top and lateral boundaries of the model domain, which were derived from the 3-hourly global model outputs for the year 2010

(http://acmg.seas.harvard.edu/geos/; Sudo et al., 2002, respectively). The global models were run by University of Tennessee (USA) and Nagoya University (Japan), respectively. GEOS-Chem was run with a 2.5º×2º horizontal resolution and 47 vertical layers and Chemical AGCM for Study of Atmospheric Environment and Radiative Forcing (CHASER) was run with a 2.8º× 2.8º horizontal resolution and 32 vertical layers. Some models made boundary conditions depending on their own previous experience.

**2.3 Observational data for O$_3$**

In this study, East Asia has been divided into four sub-regions as shown in Fig.1. The selection of the sub-regions is based on emissions, climate and observation data coverage. The North China Plain (EA1), Yangtze River Delta (EA2), and Pearl River Delta (EA3) represent the highly industrialized regions in the mid-latitudes. EA1 and EA2 have a temperate and tropical continental monsoon climate

with marked seasonality, respectively. EA3 is located in the south of China, and is less affected by the continental air masses. EA4 consists of the northwest Pacific and Sea of Japan, and represents the downwind regions of Asian continent with a marine climate.

Hourly O$_3$ and NO$_x$ observations in the year 2010 in East Asia were obtained from CERN, PRD-RAQMN), and EANET. The CERN was built by the Institute of Atmospheric Physics, Chinese Academy

of Sciences and consists of 21 surface stations within an area of 500 × 500 km$^2$ in North China Plain (EA1 sub-region; Ji et al., 2012). These stations were set up according to the United States Environmental Protection Agency method designations. The PRD RAQMN was jointly established by the governments of the Guangdong Province and the Hong Kong Special Administrative Region and consists of 16 automatic air quality monitoring stations across the EA3 sub-region (Zhong et al., 2013). Thirteen of

these stations are operated by the Environmental Monitoring Centers in Guangdong Province and the other three are located in Hong Kong and are managed by the Hong Kong Environmental Pollution Department. The EANET was launched in 1998 to address acid deposition problems in East Asia, following the model of the Cooperative Program for Monitoring and Evaluation of the Long-Range



Transmission of Air Pollutants in Europe. In this study, seven stations in the northwest Pacific and Japan (EA4 sub-region) were selected for use in evaluating the model performance level in the downwind regions of the Asian continent. More information on the EANET can be found at http://www.eanet.asia/. Note that only stations with at least 75% data validity were chosen.

5  ## 3. Model validation and general statistics

### 3.1 Annual concentrations of surface $O_3$, nitric oxide (NO) and nitrogen dioxide ($NO_2$)

Fig.2 provides a concise comparison of model performance on annual $O_3$, NO and $NO_2$ in three sub regions in East Asia. A box-and-whisker representation was used to show the frequency distribution of monthly concentrations at stations in each sub-region. Note that because the Yangtze River Delta has 10  only one station, a comparison of stations in Yangtze River Delta (EA2) is not shown in Fig. 2. In general, the majority of models significantly overestimated annual surface $O_3$ compared with the observations in EA1, EA3 and EA4. M11 in sub-regions EA1 and EA3, M7 in EA3 and EA4 and M8 in all sub-regions were exceptions. M11 simulated $O_3$ in EA1 and EA3 agreed with observations, achieving a root mean square error (RMSE) of 9.5 parts per billion volume (ppbv) and 13.3 ppbv (Table 2). The simulation of 15  $O_3$ in M7 was close to observations in EA3 and EA4. Interestingly, M8 underestimated $O_3$ in all regions, which was opposite to other models.

The performance levels of models for simulating $O_3$ were closely related to their performances for $NO_2$ and NO. In highly polluted regions (EA1 and EA3), a persistent underestimation of NO was evident across most models. An exception was M8, which overestimated NO mixing ratios in all sub-regions by 20  40-50%. This indicated M8 had the strongest $O_3$ titration that resulted in lower $O_3$ than other models and observations. An interesting phenomenon was that M7 performed better at simulating $O_3$ than most other models did, although its performance at modeling NO was comparable to other models in EA1 and EA3. Therefore, the intercomparison of $NO_x$-$O_3$ chemistry between M7 and other models is needed in the next work. In EA4, all models but M8 showed a consistent performance with respect to NO and $NO_2$, although 25  their performance regarding $O_3$ varied greatly. This suggests that $O_3$ was significantly affected by other factors in addition to local chemistry in EA4.





### 3.2 Monthly variation of surface $O_3$, NO and $NO_2$

Fig. 3 presents the monthly mean concentrations of $O_3$, NO and $NO_2$ in four sub-regions over East

Asia. All models captured the observed seasonal cycles of $O_3$, NO and $NO_2$ in EA1. Overestimates of $O_3$

of 30-60 ppbv (out of total observed values of 20-40 ppbv) in May-September were found in most models

except M11. In the same period (May-September), simulations of NO and $NO_2$ by these models appeared

to be consistent with observations, attaining mean biases of < 10 ppbv. This suggests that the

intercomparison on $O_3$ production efficiency per $NO_x$ in these models is needed. The M11 achieved the

best model reproductivity of monthly mean $O_3$ in EA1. The largest model bias and intermodel variability

for NO and $NO_2$ appeared in winter. These model biases likely came from the $NO_x$ surface emissions,

dry deposition, vertical diffusion and heterogeneous chemistry, which will further be discussed in next

section. In EA3, most models (except M7, M8 and M11) exhibited a two-peak seasonal cycle but the

observation exhibited a one-peak seasonal cycle. The $O_3$ concentration in January-May was significantly

overestimated by 15-35 ppbv (out of observed values of 20-30 ppbv). In other months, $O_3$ was slightly

overestimated by these models (~10 ppbv). The underestimation of NO titration strength partly explained

the overestimation of $O_3$. Simulation results for NO fell in the range of 1-5 ppbv in most models, which

was much less than observed concentration (10-25 ppbv). M11 captured the observed January-May $O_3$

because of relatively high NO concentrations. However, NO was overestimated by M11 in May-

September, which led to the underestimation of $O_3$. M7 seems to have achieved the best reproducibility

for $O_3$, but its simulated values of NO were only 10-30% of observations. In EA4, spatially averaged $O_3$

concentrations often differ by more than 20 ppbv in the individual models. The highest intermodel

variability on $O_3$ appeared in May-October, which overestimated $O_3$ in comparison to observations by

10-40 ppbv. Similar results have been found in MICS-Asia II and other model inter-comparison project

under the Task Force on Hemispheric Transport of Air Pollution (TF HTAP), which suggested that such

results may stem from the difference in the representation of dispersion by southwesterly clean marine

air masses in different metrological fields used in CTMs (Han et al.,2008; Fiore et al., 2009). In this study,

however, most model employed the common reference meteorological fields. This indicated the

representation of regional photochemical chemistry seems to be responsible for the model intermodel

variability and overestimation rather than the representation of southwesterly winds. Interestingly,




although M8, M9 and M14 exhibited a similar magnitude with observations in June-September, they significantly underestimated observations in other months by 200-300%. A detailed investigation is required in future studies. In contrast to $O_3$, the simulated $NO_2$ results in the models exhibited a satisfactory consistency, and agreement with observations. 4) As shown in Fig.1, only 1 station exists in EA2. Thus, the model validation with observations was likely beset by large uncertainties in EA2. Most models reproduced the $O_3$ general seasonal cycle with May-October maximum-winter minimum (Figure not shown). In May-October, only M7 and M11 estimated $O_3$ concentrations at the same magnitude as measurements, and other models overestimated $O_3$ by 100-200%. In winter and spring (November-April), half participant models (M1, M4, M7, M11, M12 and M14) agreed well with observation well, and others overestimated observations by 50%-100%. As other sub-regions, M8 underestimated $O_3$ for the whole year in EA2 because it overestimated NO by 300%. Most models appeared to have difficulties in capturing the observed NO concentrations and exhibited large scatter effects in winter. Two exceptions were M5 and M11. M5 and M11 achieved satisfactory performances in summer and other seasons. For $NO_2$, most models (except M13 and M5) showed a good consistency with observations, and a lower bias than for $O_3$ and NO.

### 3.3   Diurnal concentrations of surface $O_3$

Sub-regional $O_3$ diurnal variations are shown in Fig.4. In general, model results for four sub-regions exhibited a larger spread with a magnitude of 10-50 ppbv throughout the diurnal cycle than that in Europe and North America (Solazzo et al., 2012). This indicated that models had difficulty dealing with $O_3$ in East Asia. In EA1, the deepest diurnal variation of observed $O_3$ appeared in summer. The majority of the models exhibited a consistent overestimation (20-60 ppbv) throughout the diurnal cycle to varying degrees, with the exception of one outlying model (M8), which systematically underestimated $O_3$ concentration. Among models, M11 exhibited the best model performance level on peak daily $O_3$ concentrations of 60 ppbv in 14:00-16:00. On nighttime, M11 had a slight overestimation of 10 ppbv, due to difficulties in dealing with vertical mixing. Compared with summer, models' performances had a significant improvement in winter because of the weak intensity of photochemical reactions, except M2, M10 and M8. Differences between observations and most simulations in both nighttime and daytime were within 5 ppbv. The contrast of the models' performances between summer and winter implied that



the variety of parametrizations on chemistry in different models partly explained the intermodel variability of simulated $O_3$ in EA1 (North China Plain).

In EA3 (Pearl River Delta, China), the majority of models agreed well with the diurnal variation in autumn. In other seasons, most models had a tendency to overestimate the $O_3$ concentrations in both

daytime and nighttime. In particular, the overestimated magnitude exceeded 10 ppbv and 25 ppbv (out of observed values of 20-35 ppbv) nighttime and daytime, respectively. M11 reproduced the observed $O_3$ in spring, but underestimated $O_3$ in summer and autumn.

In EA4, all models captured the small diurnal variation of $O_3$ in four seasons. However, significant intermodel variability still existed throughout the year. As shown in Fig.4, the amplitude of intermodel

variability except M8 and M14 reached approximately 20 ppbv and approximately 10 ppbv in spring-summer and autumn-winter, respectively. Compared with the observations, the majority of models except for that of M8 and M14 generally reproduced the magnitude of observed values in spring and winter. Observations lay in the middle of simulated values. In summer, the majority of models overestimated observed both daytime and nighttime $O_3$. As discussed in section 3.2, M8 and M14 exhibited the lowest

$O_3$ among models in the whole year.

### 3.4   Error statistics

In this section, we present statistics concerning the performance levels of the models. On a yearly basis, all models showed the  highest (0.8-0.9) and lowest (0.1-0.6) correlation coefficients for $O_3$ in EA1 and EA3, respectively (Table 2). The high correlations in EA1 were mainly because the summer-

maximum and winter-minimum seasonal cycle is the typical pattern in polluted regions that were well represented in all the participating models. The large overestimation of most models in May-September led to high normalized mean bias (NMB:0.25-1.25) and RMSE (10-33 ppbv) in EA1. M11 had the lowest NMB (0.09) and RMSE (9.46 ppbv) among models. So, the model intercomparison between M11 and other models is helpful for improving the model performance level in in EA1. In EA3, M9 and M10 had

larger correlations than the other models. However, their NMB and RMSE were also the highest. This implied that systematic model biases existed in these two models. The positive bias in the majority of models except M7 and M11 was mainly caused by a large overestimation during the winter and spring seasons. M7 exhibited a lower NMB and RMSE than other models, but its correlation was only 0.29.



M11 underestimated $O_3$ concentrations by 25%. Investigating differences of model parameterization between M7, M11 and others is a good way to improve the model performance level in this region. In EA4, the correlations exhibited the largest intermodel variability among all sub-regions, ranging from -0.13-0.65. M7 showed the lowest NMB and RMSE. This is likely caused by the cancelling effect of its overestimation in summer and underestimation in other seasons (Fig.3).

For NO, correlations of models in EA1 ranged from 0.57-0.68, which indicated all models did a good job in reproducing the spatial variability of NO in this sub-region (Table 3). The NMBs indicated underestimation by models except M8 which mostly occurred in winter (Fig.3). This underestimation partly was attributed to the coarse model horizontal resolution (45km) used in the MICS-Asia III, which hardly reproduced concentrations of short-lived species. Although most of the models employed the same emission inventory and meteorological field, EA1 still had a high model intermodel variability (scattered NMBs). This implied that the treatment of models on chemistry, vertical diffusion and dry deposition may have contributed to this underestimation of NO. In contrast to most models, M8 overestimated NO concentrations in all three sub-regions. It is noted that observations of NO were too low (<0.3 ppbv) in EA4 to be discussed in this study.

Table 4 shows the statistics of models' performance levels for $NO_2$. In general, most models exhibited a better performance levels for representing $NO_2$ than $O_3$ and NO in EA1. The NMBs ranged from -0.28-0.32, which were much lower than $O_3$ (0.48-1.25). The correlations were 0.54-0.66, implying the reliable model performance levels for reproducing the spatial and month-to-month variability of $NO_2$ in EA1. Similar to $O_3$ and NO, the correlation coefficients of $NO_2$ in EA3 remained low. Thus, a dedicated investigation on $O_3$, NO and $NO_2$ in EA3 is urgent, but beyond the scope of this study. In EA4, correlation coefficients ranged from 0.5-0.72. The NMBs and RMSEs except M8 ranged from -0.42-0.46 and 0.91-1.79 ppbv, respectively.

Last but not least, large intermodel variability for $O_3$, NO and $NO_2$ occurred in EA1, EA3 and EA4, higher than for Europe and North America despite, using the same meteorological fields and emissions data (Solazzo et al., 2012). This indicated that treatment of models' parameterizations on physical and chemical processes contains nonnegligible uncertainties in East Asia. We must thus investigate the possible causes and improve the model's performance levels for $O_3$ in East Asia.



### 4. Investigation of intermodel variability on O₃

In MICS-Asia II, Han et al. (2008) briefly attributed the intermodel variability to the diversity of meteorological fields, emissions, boundary conditions, model treatment of chemistry, vertical diffusion and dry deposition. Because every model in MICS-Asia II employed their own input data, these potential reasons were not carefully examined. In MICS-Asia III, the postprocess to the common reference input data set maybe caused some discrepancies between models, because they have their own vertical structures. In addition, three models applied their own meteorological fields, which were different from the meteorology employed by other models. Thus, we compared the PBLH, emissions fluxes, dry deposition velocities and relationships between $NO_x$ and $O_3$ in the sub-regions, as well as the vertical profiles of $O_3$ and its precursors among models.

#### 4.1 Daytime PBLH

The evolution of the PBLH plays a major role for the $O_3$ and its precursors. In general, $O_3$ precursors are mostly constrained within the boundary layer (Quan et al., 2013). A better understanding of the evolution of PBL is essential for the interpretation of model biases and intermodel variability. Fig.5 presents the monthly variations of spatial mean daytime PBLH (08:00-18:00 LST) in M1, M4, M7, M8 and M11 at observed stations in EA1, EA3 and EA4. These models were selected because their simulations are largely scattered on $O_3$ and its precursors and covered the overall variability of all the models in this investigation. In EA1, all the selected models exhibited the spring-maximum and winter-minimum season cycle, which captured the major pattern of observations (Guo et al.,2016). In the climatology of PBL derived from the radiosonde by Guo et al. (2016), daytime PBLH in EA1 (North China Plain) ranged from 0.5 km in winter and 1.5 km in spring. The magnitudes of simulated PBLHs were also consistent with observations. Among models, the simulated PBLHs were very close between M1, M4, M7 and M11. The simulated PBLH by M8 was systematically higher than those by other models, but the positive bias was less than 100-150 m (<10-20%). In EA3, larger scatters of PBLH appeared than in EA1, which were almost exclusively caused by the difference between M8 and other models. The variability between M1, M4, M7 and M11 was only approximately 50 m. Compared with the climatological observations, the simulations remained at a similar magnitude with the radiosonde data




(Guo et al.,2016). In EA4, all models exhibited the winter-maximum pattern of PBLH, which was consistent with those derived from European Centre for Medium-Range Weather Forecasts Reanalysis Data (Engeln et al., 2013). In particular, PBL in May-October, the season with the highest intermodel variability of $O_3$, was quite consistent between models (<50 m). This consistency of models on PBLH in these sub-regions implied that PBL hardly explained the large intermodel variability and model biases in East Asia.

### 4.2    Emissions

In the Phase III of MICS-Asia project, the anthropogenic emission inventory in all models basically came from the monthly gridded MIX inventory at 0.25°× 0.25° resolution (Li et al.,2016). The mapping of MIX onto the different  model grids and different months could lead to some discrepancies between models, which can cause an intermodel variability in concentrations of pollutants.  Fig.6 presents the spatial averaged monthly NO emission fluxes of M1, M2, M4, M7, M8 and M11 at stations over each sub-region. In general, two groups can be formed: one consisting of M1, M8 and M11, and the other consisting of M2, M4 and M7. NO emissions in the two groups were consistent in EA1, with magnitudes of around 0.8 $\mu g/m^2/s$ and 0.6 $\mu g/m^2/s$, respectively. Interestingly, the simulated $NO_2$, NO and $O_3$ evenly presented a high intermodel variability in the same group. For example, the highest (M1) and lowest (M8) values of simulated summer $O_3$ in the first group were 80 ppbv and 30 ppbv, respectively, with the same vertical structure (Fig.3). Fig. 6 clearly indicates that the difference in emissions allocations contributed to the simulation variability. In the future, the projected gridded anthropogenic emissions should be provided to each group to eliminate the possibility that each group uses different mapping method.

### 4.3   Dry depositions

Previous studies revealed that dry deposition processes are the key sink of $O_3$, accounting for about 25% of total removed from the troposphere (Lelieveld and Dentener, 2000). The uncertainty of dry deposition in CTMs is still high because many processes are heavily parameterized in models (Hardacre et al.,2015). In East Asia,the land cover is highly heterogeneous, which brings additional difficulties to the simulation of dry deposition. The surface cover class in EA1 is the most complex, and includes deciduous broad-leaf forest, urban and cropland areas. EA3 and EA4 consist of urban, ocean and islands.



In this study, the simulated dry deposition velocities of $O_3$ were compared. Simulated deposition velocities were calculated from Eq. (1):

$$V_d = F/C \qquad\qquad (1)$$

Where $F$ and $C$ represent the simulated dry deposition flux and surface $O_3$ concentrations, respectively. We determined the spatial mean dry deposition velocities at stations in each sub-region.

Fig. 7 presents the monthly spatial mean dry deposition velocities of $O_3$ in eight models over EA1, EA3 and EA4. In EA1, $O_3$ dry deposition velocities in M1, M2, M4 and M6 presented a sharp increase from July to September, ranging from 0.2 cm/s to 0.4 cm/s. The peaks of dry deposition velocities in M11, M13 and M14 were broader and extended from April to September, with a constant magnitude of 0.3-0.35 cm/s. The seasonality in M12 was small and remained at 0.1 cm/s in the whole year. The lower dry deposition velocities of $O_3$ from M1, M2, M4 and M6 than that of M11 partly explained higher summer surface $O_3$ from those simulations than that from M11. However, M13 and M14 still produced high $O_3$ concentrations in May-September although their dry deposition velocities were similar to that of M11(Fig.3). This suggested that there were other factors besides dry deposition playing important roles in the overestimation of summer $O_3$ in the majority of models. The intermodel variability between models were expected. In MICS-Asia III, M1, M2, M4 and M6 are the same model with different versions. Hence the dry deposition velocities were consistent among these models. M11, M12, M13 and M14 employed their own vertical structures or meteorological drivers, which partly contributed to differences in $O_3$ dry deposition velocities as compared with M1, M2, M4 and M6. Interestingly, deposition velocities simulated in M11, M13 and M14 were quite similar. As for M12, the unique dry deposition parameterization (Zhang et al., 2001) was believed to contribute to the intermodel variability.

In EA3, similar features with EA1 are found. M1, M2, M4 and M6 were quite consistent with each other, with a seasonal cycle of spring minimum. M11, M12 and M14 had no obvious seasonal variability, with a magnitude of 0.1-0.2 cm/s. The seasonal pattern in M13 was considerably different from the other models, exhibiting a maximum in April-September with higher dry deposition velocities (0.5 cm/s). The performance of the models for dry deposition velocities was not always consistent with $O_3$ concentrations. For example, $O_3$ concentrations in M13 still remained high levels under higher dry deposition velocities conditions.





In EA4,all but M12 simulated small dry deposition velocities of 0.02-0.04 cm/s. This was expected because stations in this region are mostly located in coastal areas and islands, and thus their results accord with the finding in Hardacre et al. (2015), who reported that the simulated $O_3$ dry deposition velocities in eighteen models in HTAP project were <0.1 cm/s over oceans. Dry deposition velocities have a considerable effect on concentrations of surface $O_3$ on oceans, although the effect in absolute terms is small. Ganzeveld et al. (2009) revealed that surface $O_3$ may differ by up to 60% when $O_3$ dry deposition velocity varied from 0.01 to 0.05 cm/s. The uncertainties on dry deposition in EA4 may contribute to the overestimation of surface $O_3$ in the majority of models, and thus more observations are needed over oceans.

### 4.4 Relationships between surface $NO_x$ and $O_3$

In general, surface $O_3$ mainly comes from the photochemistry involving $NO_x$ and VOCs in polluted regions. Theoretical and simulation results showed that $O_3$ production increased almost linearly with the $NO_x$ increase under $NO_x$-sensitive conditions and remained relatively unchanged or even decreased in $NO_x$ saturated (often called "VOCs-limited") conditions (Kirchner et al.,2001; Sillman and He et al., 2002; Tang et al., 2010). Recent observations found that regional $O_3$ in the North China (EA1) and Pearl River Delta (EA3) was changing from $NO_x$-limited to $NO_x$-saturated regions (Jin et al., 2015). Examining the $O_3$-$NO_x$ relationships is a good way of investigating sources of intermodel variability and model errors concerning on $O_3$ in East Asia. Fig.8 presents the $O_3$ concentrations as a function of $NO_x$ in May-September based on the monthly daytime (8:00-20:00) mean observed and simulated results at stations shown in Fig.1.

In EA1 (North China Plain), observations clearly revealed that $O_3$ concentrations decreased with the increase in $NO_x$ concentration. $O_3$ concentrations mostly remained high levels (40-60 ppbv) when $NO_x$ was less than 20 ppbv. This implied that $O_3$ was under $NO_x$-saturated conditions in EA1 in May-September, which was consistent with Jin et al. (2015). The 13 models showed a high intermodel variability in relationships between $O_3$ and $NO_x$. Only M5, M7, M8 and M11 showed a negative slope between $O_3$ and $NO_x$. M7 and M11 were in relative agreement with observations, reasonably. M8 showed a systematic underestimation of observed $O_3$ in the all range of $NO_x$. By contrast, M5 systematically overestimated $O_3$ concentrations, which reached 80-100 ppbv under low $NO_x$ conditions (10-20 ppbv).




Relationships between $O_3$ and $NO_x$ in M1, M2, M4, M6, M9, M10 and M14 were consistently scattered, and had no relevance to $NO_x$. Interestingly, M13 maintained a similar $O_3$ level at all $NO_x$ levels, which was different from other models and previous theoretical results.

In EA3 (Pearl River Delta), M1, M2, M4 and M6 reproduced observed $O_3$ in low $NO_x$ (< 30 ppbv) but failed to capture the low $O_3$ under high $NO_x$ conditions (30~40 ppbv). This explained the overestimation of these models for $O_3$ in May-September. By contrast, M8 and M11 produced excessively high $NO_x$ values, which resulted in their underestimation for $O_3$. In M13 and M14, $O_3$ concentrations were nearly constant in all levels of $NO_x$. $O_3$ was positively correlated with $NO_x$ in M9 and M11, which is in contrast to observations. This suggests that more attention is needed when policy-makers designate the $O_3$ regime (VOCs-limited or NOx-limited regimes) using M9, M11, M13 and M14.

Stations in EA4 are mostly located over clean oceans or islands. $NO_x$ concentrations were less than 3 ppbv, which indicated the local chemistry appeared to not be a key factors of $O_3$ formation. Thus, we did not discuss the simulated $O_3$-$NO_x$ relationship further in this study.

High intermodel variability among 13 models in the $O_3$-$NO_x$ relationship existed over polluted regions in the MICS-Asia III. In some cases, the $O_3$ regime among models was even contradictory. This suggests that more attention must be paid to the development of abatement strategies in East Asia.

### 4.5 Vertical profiles of $O_x$ ($O_x$=$NO_2$+$O_3$)

Previous studies revealed that vertical mixing of $O_3$ and its precursors can influence the ground-level $O_3$ concentrations because of the turbulent mixing and different intensities of NO titration effects in the surface and residual layer (Zhang et al., 2009). Field campaigns showed that $O_x$ was an ideal index to reflect the impact of physical transport, excluding the impact of local NO titration (Wang et al.,2006). Fig. 9 presents the vertical profiles of simulated $O_x$ in East Asia in summer and winter. A large intermodel variability of $O_x$ above 300 hPa is evident in all sub-regions, which is attributable to the various different top boundary conditions among models. For example, the lower $O_3$ mixing rations in 100-300 hPa in M2 came from its the default top conditions. As shown in Fig.9, this large variability was not transmitted to middle troposphere (400-600 hPa), in which $O_x$ concentrations were consistent among models.





In the lower troposphere, differences among models in winter were generally less than those in summer. A small variability in winter appeared below 900 hPa in three sub-regions, and slowly decreased with height. This was likely caused by the near-ground chemistry or long-range transport. One exception was M4, which significantly underestimated $O_x$ from surface to 500 hPa in EA1 and EA4, compared with other models. This suggested that vertical convection and turbulent mixing in M4 were unique compared to other models.

With the increase of solar radiation and air temperature, vertical profiles were more scattered in the lower troposphere in summer. In polluted regions (EA1), various vertical structures of $O_x$ were found below 700 hPa. $O_x$ concentrations slowly increased with height in M8 and M11, but they mixed well in the PBL and decreased from 800 hPa to 700 hPa in the other models. This discrepancy between M8, M11 and other models was presumably caused by a series of factors. One was associated with PBL schemes in models. Bank et al. (2016) pointed out that non-local and local schemes in models significantly affected the vertical structure of trace gases. Another could be related to the model performance levels for simulating $O_3$ photochemical production rates. As discussed in section 4.4, the majority of models produced more surface $O_3$ in the same $NO_x$ concentration than M8 and M11, which resulted in the accumulation of high $O_3$ in PBL in these models. In EA3, vertical structures of $O_x$ among models were consistent, but concentrations differed more than those in EA1. This is likely related to the treatments of convection and cloud activity among models. EA3 is located in subtropics, and frequent convective and cloud activity redistributed $O_x$ on the vertical dimension by strong vertical transport and changing photolysis rates.

## 5. Multi-model ensemble $O_3$ and comparison with MICS-Asia II

### 5.1 Ensemble $O_3$ at stations

Studies have demonstrated that the ensemble model usually exhibits a superior performance on $O_3$ than any single model (Solazzo et al.,2012). Table 2-4 also presents the statistics of two muti-model ensembles (Mean and Median) on $O_3$, NO and $NO_2$ in EA1, EA3 and EA4. Clearly, the $O_3$ NMB and RMSE of ensemble mean were significantly less than the ensemble median in most situations, which indicated the ensemble mean presented a better performance level to represent the observed $O_3$. Therefore,



we only presented the results of multi-model mean ensemble (ENSE). In general, ENSE performed a

better performance level than individual models for representing $NO_2$ in East Asia, reproducing the

observed seasonal cycle and magnitudes (Fig. 3). However, ENSE did not always exhibited a superior

performance for $O_3$ over certain individual model in East Asia, which was in contrast to its performance

in Europe (Fig.3). M11 and M7 agreed well with observations in EA1 and EA3, while ENSE tended to

overestimate $O_3$ concentrations in May-September in EA1 and January-September in EA3. Loon et al.

(2007) indicated that ENSE exhibited a superior performance level only when the spread of ensemble-

model values was representative of the uncertainty of $O_3$. This indicated that most models did not reflect

this uncertainty or missed key processes in MICS-Asia III.

**5.2   Spatial distribution of single model and multi-model ensemble $O_3$**

Fig.10 presents the predicted spatial distribution of seasonal averaged surface $O_3$ concentrations in

summer for individual models. All models similarly predicted the elevated $O_3$ concentration belt in the

middle-latitudes (30°-45°N). However, the magnitude of the enhanced $O_3$ were different among the

models. M5 predicted the highest $O_3$ concentration of 60-90 ppbv in this belt, whereas M8 predicted the

lowest 35-50 ppbv.  The models also consistently simulated low $O_3$ concentrations in low-latitudes (0°-

15°N), but varied by approximately 20 ppbv among models. M8, M10 and M11 showed the lowest value

(10-15 ppbv), and M2, M7 and M9 showed the highest (30-45 ppbv). The rest are consistently in a middle

range (15-25 ppbv). This discrepancy in low latitudes among models might have resulted from the

diversity of boundary conditions and dry deposition velocities. The lateral boundaries in M7, M8 and M9

came from the default configurations of models, while the rest utilized the output from CHASER, GEOS-

Chem, or MOZART-GOCART global models. The largest differences among models appeared in the

North China Plain (EA1) and its outflow pathways including Bohai Sea, East China Sea, Korea, Japan

and the Sea of Japan. Most models -M1, M2, M4, M5, M6, M10 and M14- predicated much higher

surface $O_3$ levels (75-85 ppbv) in North China Plain (EA1), where observations of 40-50 ppbv were

reported. SAPRC99 chemical mechanism used in these models except M14 partly contributed to the

overestimation. Previous studies revealed that SAPRC-99 predicts higher concentrations than CB05 and

CB4 used in other models (Luecken et al., 2008). $O_3$ concentrations decreased with the increase of

outflow distances, and reached ~60 ppbv in Sea of Japan in these models. In M8 and M11, $O_3$





concentrations were lower (30-50 ppbv) in source regions than other models (EA1) and increased in the long-range transport to Japan. This inconsistency among models have resulted from the combined influence of a series of factors that included the diversity in condensed gas-chemical mechanism and heterogeneous chemistry. Olson et al. (1997) indicated that a significant difference could appear among

models with respect to the concentrations of $O_3$ because of differences in simulated photolysis rates, specific chemical reaction rates, and various treatments of VOCs. Li et al. (2015) found that the chemical production was the dominated controlling factor of $O_3$ along the outflow pathways near the North China Plain in summer, rather than lateral and top boundary conditions. The heterogeneous chemistry largely reduced surface $O_3$ in polluted regions of China with high aerosol loadings (Li et al.,2018). Interestingly,

overhang of 30 ppbv contour lines extending into Northwest Pacific in the Asian continent outflow plume differed considerably among models. The plume of 30 ppbv or higher $O_3$ in M1-M6, M13 and M14 reached further south and east of Japan (135°E, 20°N), than M8, M10 and M11 (120°E, 30°N). In MICS-ASIAII and HTAP, differences of frequency of marine air masses from the western Pacific Ocean were thought to be possible cause of $O_3$ discrepancy over ocean among models because of different

meteorological drivers (Han et al., 2008). As discussed in section 4.1, models in this study employed the same or similar meteorological fields. This indicated that the chemistry during the long-range transport of pollutants in continental outflows seems to be a key factor causing $O_3$ simulation discrepancies. In winter, the distribution patterns of $O_3$ were quite alike among models, with high concentrations over parts of western China, northeastern India and the western Pacific from the East China Sea to south of Japan

(Fig.11). Considerably high consistency was found among models in winter compared with summer. All models predicted the low concentration in eastern China because of the titration effect of high $NO_x$ concentrations. However, the magnitude of $O_3$ were different. M5 and M8 notably underpredicted $O_3$ (~10 ppbv) than other models (15-30 ppbv). In spring and autumn (Fig.S1 and Fig.S2 in the supplements), $O_3$ concentrations were generally higher than in winter in the whole model domain because of the

enhancement of solar radiation or stratosphere-troposphere exchanging fluxes of $O_3$. A major feature consistently produced by all models was the enhancement of $O_3$ over southern Tibet, northeastern India and the western Pacific, which was generally similar to that in winter. The position of $O_3$ enhancement further north of Japan was comparable with winter.





The spatial distributions of MICS-Asia III ensemble mean surface $O_3$ (ENSE) and the coefficient of variation (CV) were presented in Fig.12. the major features in the four seasons discussed in the preceding paragraph are more clearly identified. The distribution of ENSE $O_3$ concentrations was much smoother than any individual model, due to data averaging. In summer, a region of $O_3$ in excess of 60 ppbv stretched across North China Plain and China East Sea, which was much higher than values in MICS-Asia II (45-50 ppbv) for the year of 2001(Han et al.,2008). In other seasons, the $O_3$ distribution shows higher $O_3$ over ocean than in eastern China, reflecting the $O_3$ titration from high $NO_x$ emissions. Due to the stratospheric injection, surface $O_3$ over Tibet plateau remained a high level in the whole year, ranging from 50 to 65 ppbv. The seasonal cycle of surface $O_3$ in ENSE in MICS-ASIA III agreed with that in MICS-Asia II, but $O_3$ levels in polluted regions were higher (Han et al., 2008).

The CV ranged from 0.1-0.6 in East Asia. The highest values were found in EA1 in winter. These high values in low-latitude western Pacific ($10^oS-15^oN$) and Indian Ocean were likely caused by the treatment of lateral boundaries in models. In MICS-ASIAIII, M7, M8 and M9 employed the default configurations of models, and the others employed outputs of GEOS-Chem/CHASER/MOZART-GOCART global model. Compared with MIC-Asia II, the CVs in Asian continent except winter remained a similar level in this study (0.1-0.3) (Carmichael et al.,2008).

### 5.3   Comparison with MICS-Asia II

In MICS-Asia II, model evaluation on $O_3$ were conducted in only sites in the western Pacific. Fig.13 presents the simulated and observed surface $O_3$ at these monitoring sites in the phase II and III of MICS-Asia project. Note that different models were employed in two phases. In general, most models captured the major distribution of $O_3$ at most sites in both MICS-Asia II and III. ENSE showed a good consistency in March and December of 2001 and 2010. The underestimation of $O_3$ in March at Japan sites (site 4: Sado-seki, site 5: Oki and site 6: Banryu) in Phase II was largely improved in Phase III. However, the surface $O_3$ at western Japan (site 4: Oki, site 5: Hedo and site 6: Banryu) were severely overestimated in July 2010 by 10-30 ppbv. This overestimation has not been found in Phase II, in which the difference with observations was approximately 5 ppbv. Rural sites in western Japan were located in the upwind regions of Japanese domestic emissions, and usually used to capture the impact of Asian continent outflows. The overestimated $O_3$ in North China Plain (EA1) in Phase III contributed a lot to the enhanced


concentrations at sites of western Japanese sites in July 2010. This indicated that the transboundary transport from the Asian continent in MCIS-Asia III was likely overestimated compared with that in MICS-Asia II.

## 6. Summary

In the MICS-Asia III framework, the evaluation and intercomparison of 13 CTMs were conducted with a wide variety of observations covering three Chinese industrialized regions and western Pacific, using long-term simulations for the year 2010. This study has focused on surface $O_3$ and its relevant species. In particular, surface $O_3$ in China was evaluated, which was absent in the previous model-intercomparison projects. Causes responsible for discrepancy of models with observations and

intermodel variability were investigated. Finally, a model ensemble was conducted and evaluated. Most models captured the key pattern of monthly and diurnal $O_3$ and its precursors (NO and $NO_2$) in the North China Plain, the Yangtze River Delta and the western Pacific Rim. However, the majority of models failed to capture the observed single peak (autumn-maximum) seasonal cycle of $O_3$ in Pearl River Delta of China, which exhibited a two-peak seasonal cycle.

Considerable difference between simulated and observed $O_3$ concentrations were found in all four subregions in East Asia. In North China Plain, the majority of models severely overestimated surface $O_3$ in May-September by 20-40 ppbv. The only exception was M8 , which underestimated surface $O_3$ by 10-15 ppbv. This overestimation systematically appeared in both daytime and nighttime. Similarly, most models had a predominate tendency to overestimate the daytime and nighttime $O_3$ concentrations in

January-May and May-October in Peral River Delta (EA3) and western Pacific rim (EA4), respectively. The monthly $O_3$ series also revealed that some models performed better than others in some subregions (for example, M11 in EA1), but this behavior was not uniform in time and space. For $NO_2$ and NO, models appeared to be more consistent with observations than $O_3$.

Large intermodel variability of $O_3$ existed in all subregions over East Asia in this study, with model

concentrations varying by a factor of 2 to 3 between different models. MICS-Asia II presented some potential reasons of variabilities among models, but did not explicitly examine the impact of these reasons. In this study, we directly investigated the diversity of PBLH, emissions, dry deposition, $O_3$ -$NO_x$



relationships and vertical profiles among models. This investigation revealed that the internal chemical parameterizations of models (gaseous and heterogeneous chemistry) heavily contributed to the large variability among models, even though the native schemes in models were similar. Dry deposition and vertical mixing also played important roles.

This study revealed that ensemble average of 13 models on $O_3$ (ENSE) did not always exhibit a superior performance to certain individual models in East Asia, which contrasted with its performance in Europe. This suggested that the spread of ensemble-model values had not represented all uncertainties of $O_3$ or most models in MICS-Asia III missed key processes. Unlike the performance level for $O_3$, ENSE demonstrated superior performance level than individual models for $NO_2$ in East Asia.

Compared to MICS-Asia II, MICS-Asia III was less prone to underestimation of $O_3$ in March at Japanese sites. However, it predicted too enhanced surface $O_3$ concentrations at western Japan in July because of its overestimation in the North China Plain, which was not the case in MICS-Asia II. This indicated that the transboundary transport from Asian continent was likely overestimated in MCIS-Asia III.

**Author contribution:**

JL, ZW and GC conducted the study design. JL, TN, BG, KY, JF, XW, QF, SI, HL, CK, CL, MZ, ZT, MK, HL contributed to modeling data. ML, JW, JK and QW provided the emission data. LK helped with data processing. HA, GC and ZW were involved in the scientific interpretation and discussion. JL prepared the manuscript with contributions from all co-authors

**Competing interests**:

The authors declare that they have no conflict of interest.

**Acknowledgements:**

This work was supported by the Natural Science Foundation of China (41620104008, 41571130034; 91544227;91744203), and National Key R&D Program of China (2017YFC0212402). This work was
partly supported by the Environment Research and Technology Development Fund (S-12) of the Environmental Restoration and Conservation Agency of Japan and the Ministry of Environment, Japan. We thank the Pearl River Delta Regional Air Quality Monitoring Network for observations in Pearl River



Delta. Dr. Kengo Sudo from Nagoya university and Prof. Rokjin J. Park provided us CHASER and

GEOS-Chem outputs for boundary conditions. This manuscript was edited by Wallace Academic Editing.

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




**Table and Figure captions:**

Table.1 Basic structures, schemes and relevant parameters of the fourteen participating models

Table. 2 Statistical analysis for surface $O_3$ in three subregions over East Asia (R: correlation coefficient; NMB: Normalized Mean Bias; RMSE: Root Mean Square Error)

Table. 3 Statistical analysis for surface NO in three subregions over East Asia (R: correlation coefficient; NMB: Normalized Mean Bias; RMSE: Root Mean Square Error)

Table. 4 Statistical analysis for surface $NO_2$ in three subregions over East Asia (R: correlation coefficient; NMB: Normalized Mean Bias; RMSE: Root Mean Square Error)

Fig.1 Model domain of models except M13 and M14 with locations of four sub-regions marked in this

study. Also show are locations of surface monitoring stations in this study. Note that the domains of M13 and M14 are shown in Fig.10.

Fig.2 Box-plots of observed and simulated annual $NO_2$(left column), NO (middle column) and $O_3$ (right column) frequency distribution by 13 models, averaged in stations over EA1, EA3 and EA4, and in time for the whole 2010 year. n represents the numbers of stations.

Fig.3 Time series of monthly $NO_2$, NO and $O_3$ simulated by all models and their ensembles(Ense), in ppbv, averaged over all observed stations in three subregions over East Asia (EA1: top row, EA3: middle row, EA4: bottom row). Observations are also shown by the black line. n represents the numbers of stations

Fig. 4 Seasonal mean diurnal cycle of surface $O_3$, in ppbv, as a function of hour, for all models and their

ensembles, averaged over all observed stations in three subregions over East Asia (EA1: top row, EA3: middle row, EA4: bottom row). Observations are also shown by the black line. n represents the numbers of stations

Fig.5 Simulated monthly daytime (08:00-20:00 LST) PBL height (m) by M1, M4, M7, M8 and M11, averaged over all observed stations in three subregions over East Asia (EA1: top row, EA3: middle row,

EA4: bottom row). n represents the numbers of stations

Fig.6 The same as Fig.5, but for NO emission fluxes on the first day in each month. M2 was also shown

Fig.7 The same as Fig.5, but for $O_3$ dry deposition velocities (Vd) of M1, M2, M4, M6, M11, M12, M13 and M14

Fig.8 Scatter plots between monthly daytime (08:00-20:00) surface $NO_x$ and $O_3$ at each station over

EA1(red), EA3(green)and EA4(blue) in May-October, for observations(obs) and models.

Fig.9 Simulated $O_x$ ($O_3+NO_2$) profiles in summer and winter of 2010, averaged over all observed stations in three subregions over East Asia (EA1: left column, EA3: middle column, EA4: bottom column).



Fig.10 Surface O$_3$ spatial distribution from 13 models for summer 2010 (unit:ppbv).

Fig.11 The same as Fig.10, but in winter 2010.

Fig.12 The ensemble mean seasonal surface O$_3$ concentrations and CV for the different seasons. CV is defined as the standard deviation of the modeled fields divided by the average, for the different seasons

5   Fig.13 The modeled and observed monthly mean concentrations of O$_3$ at EANET sites in the phase II (left panel) and III (right panel) of MICS-ASIA project. Solid line represents ensemble mean. Note that data in MCIS-ASIA II and III are in the period of March, July and December of 2001 and 2010, respectively. ID of Monitoring sites represents: 1: Rishiri(45.12$^o$N, 141.23$^o$E), 2:Ogasawara(27.83$^o$N, 142.22$^o$E), 3: Sado-seki (38.23$^o$N, 138.4$^o$E), 4: Oki (36.28$^o$N, 133.18$^o$E), 5: Hedo(26.85$^o$N,128.25$^o$E), 6:

10   Banryu (34.67$^o$N,131.80$^o$E)



Table1 Basic structures, schemes and relevant parameters of the fourteen participating models

| Models | M1 | M2 | M3 | M4 | M5 | M6 | M7 | M8 | M9 | M10 | M11 | M12 | M13 | M14 |
|---|---|---|---|---|---|---|---|---|---|---|---|---|---|---|
| Domain | Ref[a] | Ref[a] | Ref[a] | Ref[a] | Ref[a] | Ref[a] | Ref[a] | Ref[a] | Ref[a] | Ref[a] | Ref[a] | Ref[a] | Global | 10°N -50°N; 80°E -135°E |
| Horizontal resolution | 45km | 45km | 45km | 45km | 45km | 45km | 45km | 45km | 45km | 45km | 45km | 45km | 0.5° ×0.667° | 45km |
| Vertical resolution | 40$\sigma_p$ levels | 40$\sigma_p$ levels | 40$\sigma_p$ levels | 40$\sigma_p$ levels | 40$\sigma_p$ levels | 40$\sigma_p$ levels | 40$\sigma_p$ levels | 40$\sigma_p$ levels | 40$\sigma_p$ levels | 60$\sigma_p$ levels | 20$\sigma_p$ levels | 40$\sigma_p$ levels | 47$\sigma_p$ levels | 15$\sigma_z$ levels |
| Depth of first layer | 58m | 58m | 58m | 58m | 58m | 58m | 29m | 58m | 16m | 44m | 48m | 27m | | 100m |
| Meteorology | Standard[b] | Standard[b] | Standard[b] | Standard[b] | Standard[b] | Standard[b] | WRF/NCEP[b] | WRF/NCEP[b] | WRF/NCEP[b] | WRF/ MERRA2[b] | Standard[b] | Standard[b] | GEOS-5 | RAMS/NCEP[b] |
| Advection | Yamo (Yamartino, 1993) | Yamo | Yamo | PPM(Collella and Woodward 1984) | PPM | Yamo | 5[th] order monotonic | 5[th] order monotonic | 5[th] order monotonic | 5[th] order monotonic | Walcek and Aleksic (1998) | Walcek and Aleksic (1998) | PPM | PPM |
| Vertical diffusion | ACM2 (Pleim,2007) | ACM2 | ACM2 | ACM2 | ACM2 | ACM2 | 3[th] order Monotonic | 3[th] order Monotonic | YSU | YSU | K-theory | FTCS (Forward in Time, Center in Space) | Lin and McElroy, (2010) | ACM2 |
| Dry deposition | Wesely (1989) | Wesely (1989) | Wesely (1989) | M3DRY (Pleim et al., 2001) | M3DRY | M3DRY | Wesely (1989) | Wesely (1989) | Wesely (1989) | Wesely (1989) | Wesedly (1989) | Wesely1989 )and Zhang et al. (2003) | Wesely (1989) | Wesely (1989) |





| | | | | | | | | | | | | | |
|---|---|---|---|---|---|---|---|---|---|---|---|---|---|
| **Wet deposition** | Henry's Law | Henry's Law | Henry's Law | Henry's Law | Henry's Law | ACM | Henry's Law | AQCHEM | Easter et al., (2004) | Grell | Henry's Law | Henry's Law | Henry's Law | Henry's Law |
| **Gas chemistry** | SAPRC99(Carter,2000) | CBM05(Yarwood et al.,2005) | SAPRC99 | SAPRC99 | SAPRC99 | SAPRC99 | RACM-ESRL with KPP | RACM (Goliff et al., 2013) | RADM2 (Stockwell et al., 1990) | RADM2 | CBMZ (Zaveri et al.,1999) | SAPRC99(Carter,2000) | NOx-Ox-HC chemistry mechanism | SAPRC99 |
| **Aqueous chemistry** | ACM-ae6 | ACM-ae5 | ACM-ae5 | ACM-ae5 | ACM-ae5 | ACM-ae5 | CMAQ simplified Aqueous chemistry | AQCHEM | Walcek and Taylor (1986) | None | RADM2 (Stockwell et al., 1990) | Walcek and Teylor (1986) Carlton et al. (2007) | - | ACM |
| **Inorganic mechanism** | AER06(Binkowski and Roselle, 2003) | AER06 | AER05 | AER05 | AER05 | AER05 | MADE (Ackermann et al., 1998) | MADE | MADE | GOCART | ISORROPIAv1.7(Nenes et al.,1998) | Kajino et al. (2012) | ISORROPIAv1.7 | ISORROPIAv1.7 |
| **Boundary conditions** | GEOS-Chem global model (Martin et al.,2002) | Default | GEOS-Chem global model | CHASER global model (Sudo et al., 2002a, 2002b) | CHASER global model | CHASER global model | Default | CHASER global model | GEOS-Chem global model | MOZART + GOCART global models[c] | CHASER global model | CHASER global model | / | GEOS-Chem global model |
| **Two-way feedback** | Off-line | Off-line | Off-line | Off-line | Off-line | Off-line | On-line | On-line | On-line | Off-line | Off-line | On-line | Off-line | Off-line |

[a] Ref represent the referenced domain by MICS-ASIA III project.

[b] Unified represents the reference meteorological field provided by MICS-ASIAIII project; WRF/NCEP and WRF/MERRA represents the meteorological field of the participating model itself, which was run by WRF driven by the NCEP and Modern Era Retrospective-analysis for Research and Applications (MERRA) reanalysis dataset.



ᵇBoundary conditions of M10 are from MOZART and GOCART (Chin et al., 2002; Horowitz et al.,2003), which provided results for gaseous pollutants and aerosols, respectively.




Table 2 Statistical analysis for surface O$_3$ in three subregions over East Asia (R: correlation coefficient; NMB: Normalized Mean Bias; RMSE:Root Mean Suqare Error)

| Models | Region | R | NMB | RMSE | Region | R | NMB | RMSE | Region | R | NMB | RMSE |
|---|---|---|---|---|---|---|---|---|---|---|---|---|
| M1 | | 0.89 | 0.52 | 19.79 | | 0.48 | 0.31 | 14.41 | | 0.57 | 0.28 | 15.49 |
| M2 | | 0.90 | 0.64 | 18.13 | | 0.10 | 0.35 | 15.06 | | 0.66 | 0.24 | 13.83 |
| M4 | | 0.87 | 0.44 | 18.78 | | 0.41 | 0.36 | 14.15 | | 0.01 | 0.05 | 17.57 |
| M5 | | 0.87 | 0.42 | 19.00 | | 0.30 | 0.14 | 13.38 | | 0.34 | 0.31 | 19.28 |
| M6 | | 0.90 | 0.88 | 25.41 | | 0.15 | 0.44 | 17.41 | | 0.52 | 0.31 | 16.52 |
| M7 | EA1(n=19)[a] | 0.84 | 0.25 | 10.03 | EA3(n=13) | 0.29 | -0.08 | 11.11 | EA4(n=8) | 0.60 | 0.02 | 10.97 |
| M8 | | 0.78 | -0.47 | 13.52 | | 0.20 | -0.59 | 19.54 | | 0.55 | -0.27 | 15.32 |
| M9 | | 0.85 | 0.59 | 14.84 | | 0.63 | 0.48 | 15.69 | | 0.26 | -0.09 | 13.27 |
| M10 | | 0.82 | 1.24 | 32.70 | | 0.51 | 0.72 | 21.71 | | 0.52 | 0.11 | 12.68 |
| M11 | | 0.81 | 0.09 | 9.46 | | 0.34 | -0.25 | 13.40 | | 0.65 | 0.15 | 12.09 |
| M12 | | 0.89 | 0.55 | 18.53 | | 0.36 | 0.30 | 13.31 | | 0.57 | 0.11 | 11.81 |
| M13 | | 0.86 | 0.95 | 22.69 | | 0.25 | 0.50 | 17.04 | | 0.63 | 0.09 | 11.04 |





|  | | | | | | | | |
| --- | --- | --- | --- | --- | --- | --- | --- | --- |
| M14 | 0.86 | 0.75 | 23.33 | 0.12 | 0.40 | 17.01 | -0.13 | -0.30 | 20.03 |
| Ensemble Mean | 0.89 | 0.53 | 15.92 | 0.38 | 0.23 | 11.76 | 0.52 | 0.08 | 11.93 |
| Ensemble Media | 0.89 | 0.56 | 17.86 | 0.37 | 0.31 | 13.29 | 0.54 | 0.11 | 12.06 |

a: n represents the numbers of observation stations




Table 3 Statistical analysis for surface NO in three subregions over East Asia (R: correlation coefficient; NMB: Normalized Mean Bias; RMSE:Root Mean Suqare Error)

| Models | Region | R | NMB | RMSE | Region | R | NMB | RMSE | Region | R | NMB | RMSE |
|---|---|---|---|---|---|---|---|---|---|---|---|---|
| M1 | | 0.58 | -0.35 | 20.68 | | 0.22 | -0.81 | 15.16 | | 0.03 | -0.35 | 0.23 |
| M2 | | 0.57 | -0.14 | 23.73 | | 0.14 | -0.73 | 15.21 | | 0.06 | -0.27 | 0.19 |
| M4 | | 0.60 | -0.61 | 22.29 | | 0.18 | -0.87 | 15.72 | | 0.00 | -0.39 | 0.20 |
| M5 | | 0.57 | -0.07 | 20.34 | | 0.24 | -0.29 | 13.80 | | 0.02 | 0.08 | 0.35 |
| M6 | | 0.60 | -0.71 | 23.36 | | 0.11 | -0.89 | 15.94 | | 0.15 | -0.70 | 0.16 |
| M7 | EA1(n=19) | 0.63 | -0.75 | 24.91 | EA3(n=13) | 0.04 | -0.78 | 15.32 | EA4(n=8) | 0.27 | -0.40 | 0.15 |
| M8 | | 0.65 | 0.91 | 26.89 | | 0.29 | 1.14 | 25.06 | | 0.24 | 3.53 | 0.94 |
| M9 | | 0.58 | -0.82 | 27.73 | | 0.32 | -0.93 | 16.72 | | 0.22 | -0.54 | 0.14 |
| M10 | | 0.63 | -0.90 | 27.97 | | 0.27 | -0.94 | 16.30 | | 0.39 | -0.51 | 0.14 |
| M11 | | 0.61 | -0.34 | 19.92 | | 0.04 | -0.05 | 14.86 | | 0.41 | 0.09 | 0.14 |
| M12 | | 0.62 | -0.55 | 21.19 | | 0.13 | -0.85 | 15.64 | | 0.17 | -0.48 | 0.16 |
| M13 | | - | - | - | | - | - | - | | - | - | - |



| | | | | | | | | | |
|---|---|---|---|---|---|---|---|---|---|
| M14 | 0.68 | -0.66 | 22.74 | 0.01 | -0.66 | 14.77 | 0.24 | -0.50 | 0.15 |
| Ensemble Mean | 0.63 | -0.42 | 20.12 | 0.21 | -0.55 | 13.58 | 0.20 | -0.03 | 0.19 |
| Ensemble Media | 0.62 | -0.58 | 21.66 | 0.17 | -0.83 | 15.40 | 0.17 | -0.45 | 0.16 |

a: n represents the numbers of observation stations



Table 4 Statistical analysis for surface $NO_2$ in three subregions over East Asia (R: correlation coefficient; NMB: Normalized Mean Bias; RMSE:Root Mean Suqare Error)

| Models | Region | R | NMB | RMSE | Region | R | NMB | RMSE | Region | R | NMB | RMSE |
|---|---|---|---|---|---|---|---|---|---|---|---|---|
| | EA1(n=19) | | | | EA3(n=13) | | | | EA4(n=8) | | | |
| M1 | | 0.59 | -0.18 | 11.08 | | 0.33 | -0.30 | 12.92 | | 0.54 | 0.27 | 1.51 |
| M2 | | 0.64 | -0.25 | 11.30 | | 0.25 | -0.43 | 14.85 | | 0.43 | -0.07 | 1.13 |
| M4 | | 0.65 | -0.28 | 11.62 | | 0.26 | -0.32 | 13.79 | | 0.56 | -0.07 | 1.04 |
| M5 | | 0.57 | 0.08 | 10.86 | | 0.30 | 0.09 | 12.91 | | 0.60 | 0.46 | 1.79 |
| M6 | | 0.65 | -0.22 | 11.04 | | 0.23 | -0.30 | 13.86 | | 0.56 | -0.23 | 0.90 |
| M7 | | 0.59 | -0.22 | 11.42 | | 0.20 | -0.25 | 13.24 | | 0.65 | 0.19 | 1.42 |
| M8 | | 0.43 | 14.32 | 11.90 | | 0.43 | 0.15 | 10.97 | | 0.72 | 2.38 | 4.46 |
| M9 | | 0.60 | 32.30 | 18.80 | | 0.51 | -0.37 | 12.66 | | 0.49 | 0.05 | 1.66 |
| M10 | | 0.61 | -10.61 | 10.65 | | 0.15 | -0.08 | 12.81 | | 0.63 | 0.06 | 1.33 |
| M11 | | 0.54 | 0.00 | 10.82 | | 0.24 | 0.13 | 13.56 | | 0.69 | 0.36 | 1.58 |
| M12 | | 0.63 | -0.16 | 10.76 | | 0.25 | -0.24 | 13.78 | | 0.61 | -0.05 | 0.91 |
| M13 | | - | - | - | | - | - | - | | - | - | - |



| | | | | | | | | |
|---|---|---|---|---|---|---|---|---|
| M14 | 0.66 | -0.12 | 10.00 | 0.08 | -0.22 | 14.50 | 0.60 | 0.42 | 0.91 |
| Ensemble Mean | 0.65 | -0.09 | 9.89 | 0.29 | -0.18 | 12.16 | 0.64 | 0.25 | 1.33 |
| Ensemble Media | 0.65 | -0.13 | 10.07 | 0.27 | -0.23 | 12.85 | 0.59 | 0.06 | 1.23 |

a: n represents the numbers of observation stations





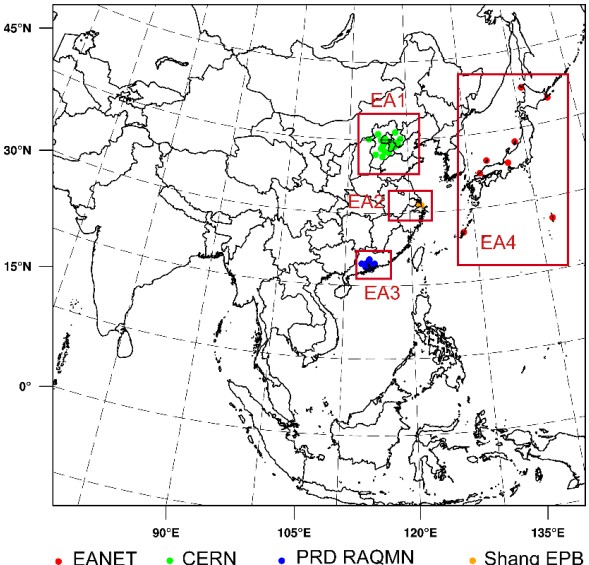

*Fig.1 Li et al., 2018*



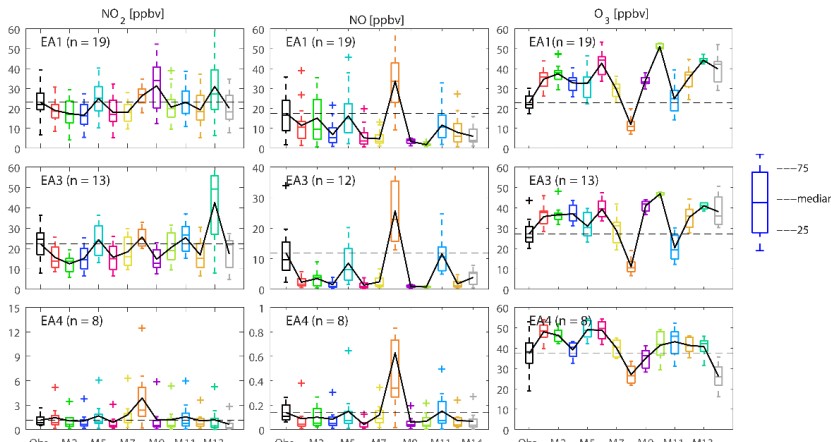

*Fig.2 Li et al., 2018*




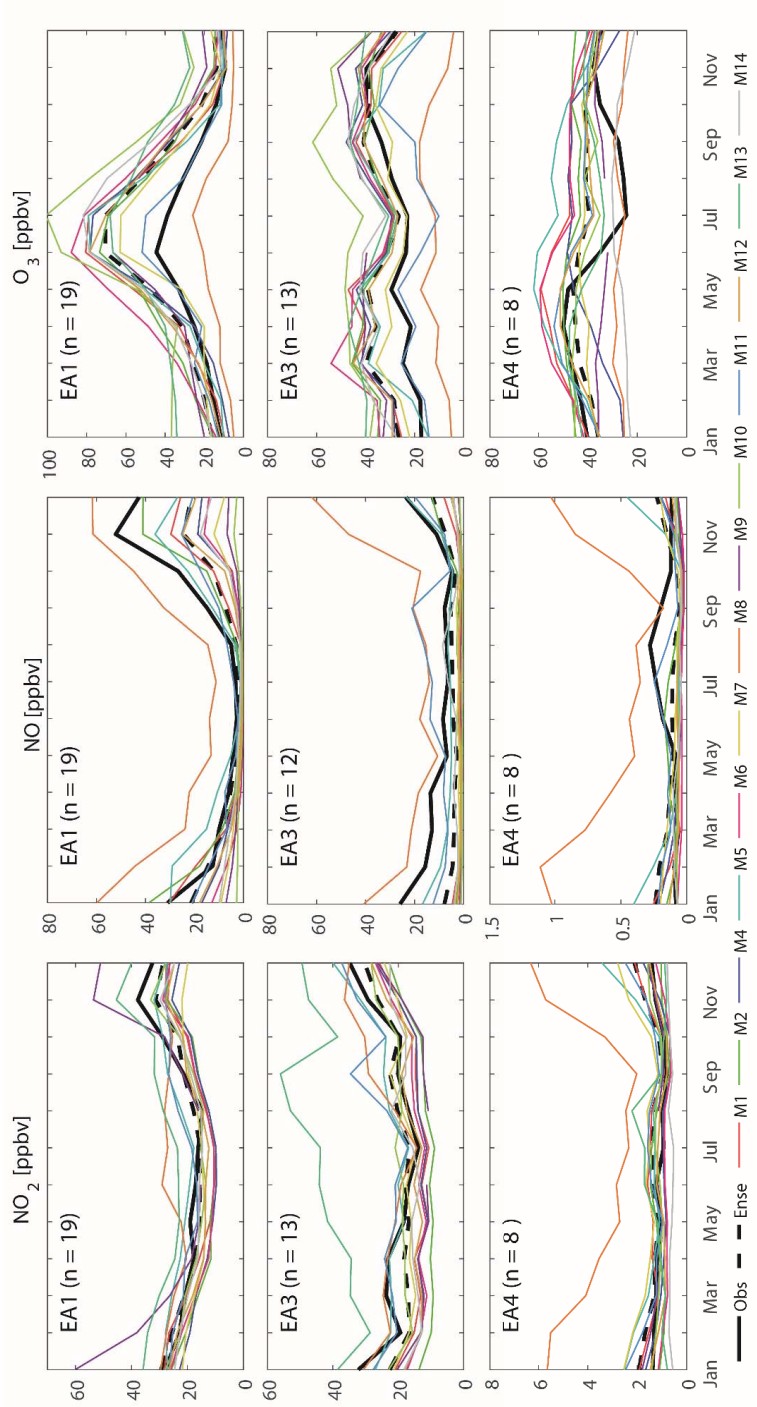

*Fig.3 Li et al., 2018*





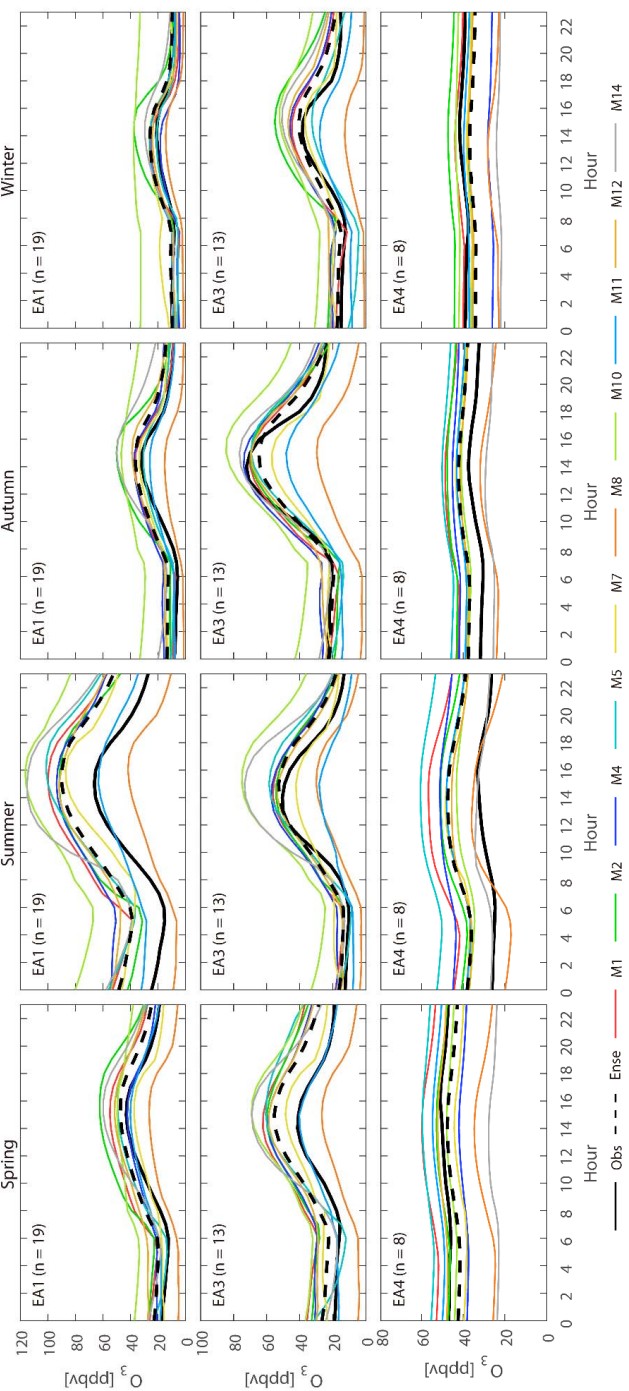

*Fig.4 Li et al., 2018*





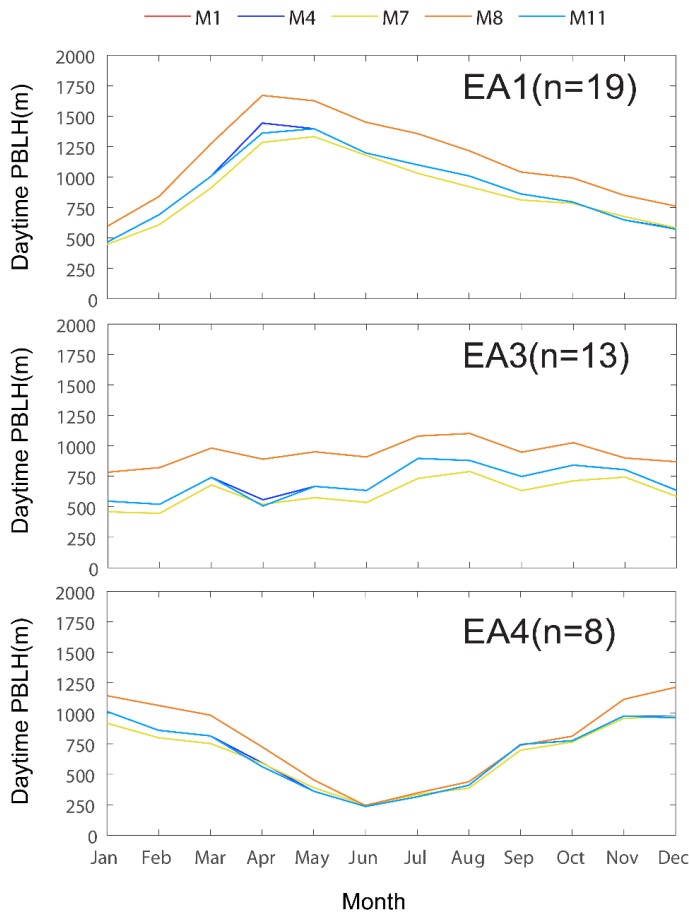

*Fig.5 Li et al., 2018*



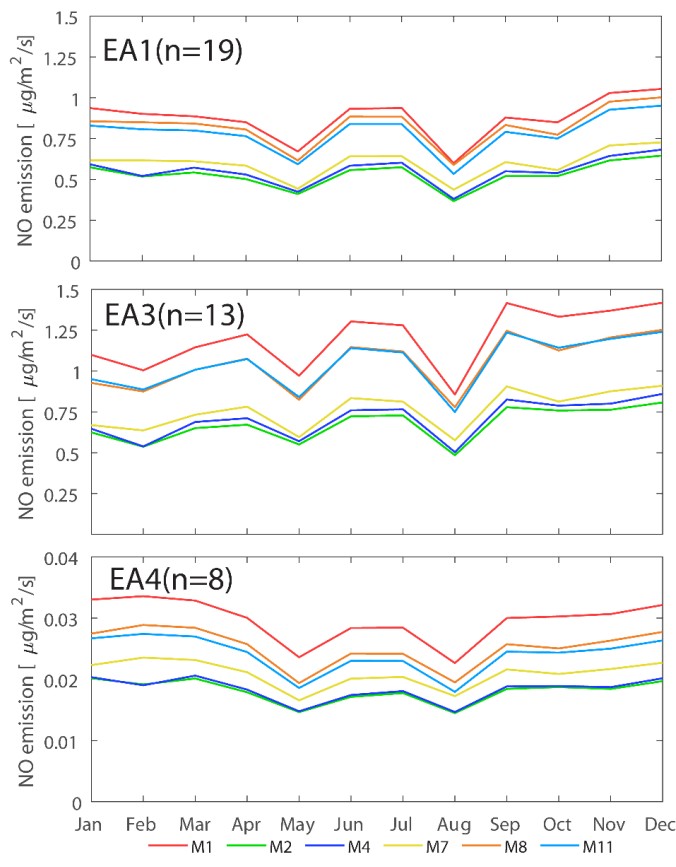

*Fig.6 Li et al., 2018*

.





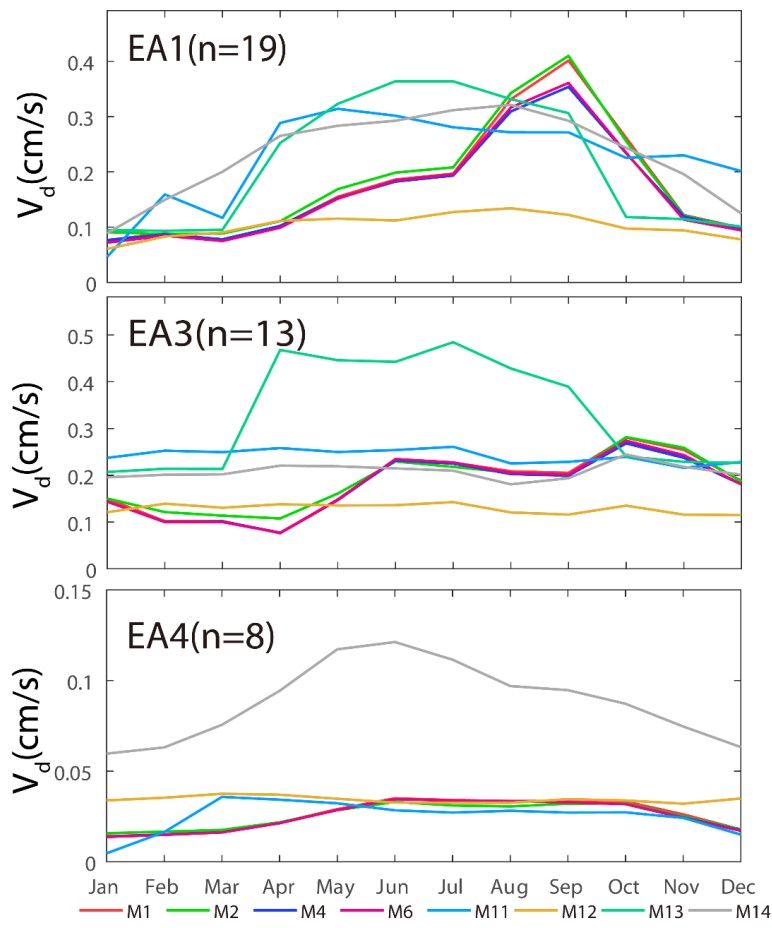

*Fig.7 Li et al., 2018*





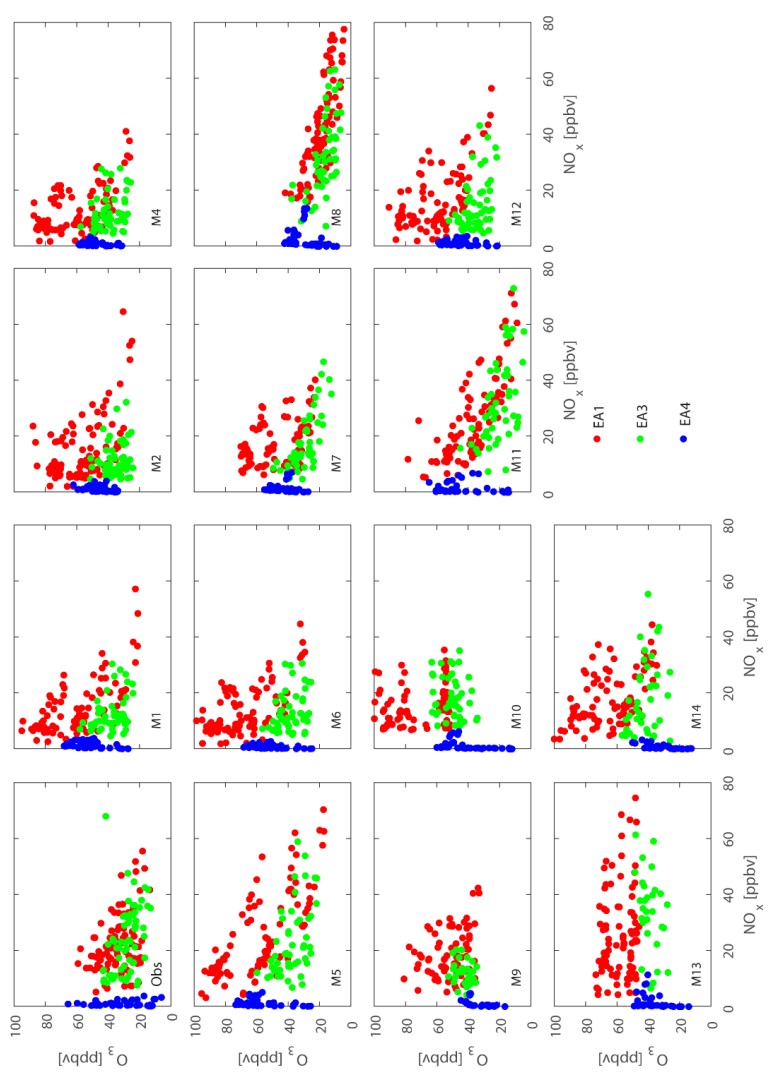

*Fig.8 Li et al., 2018*





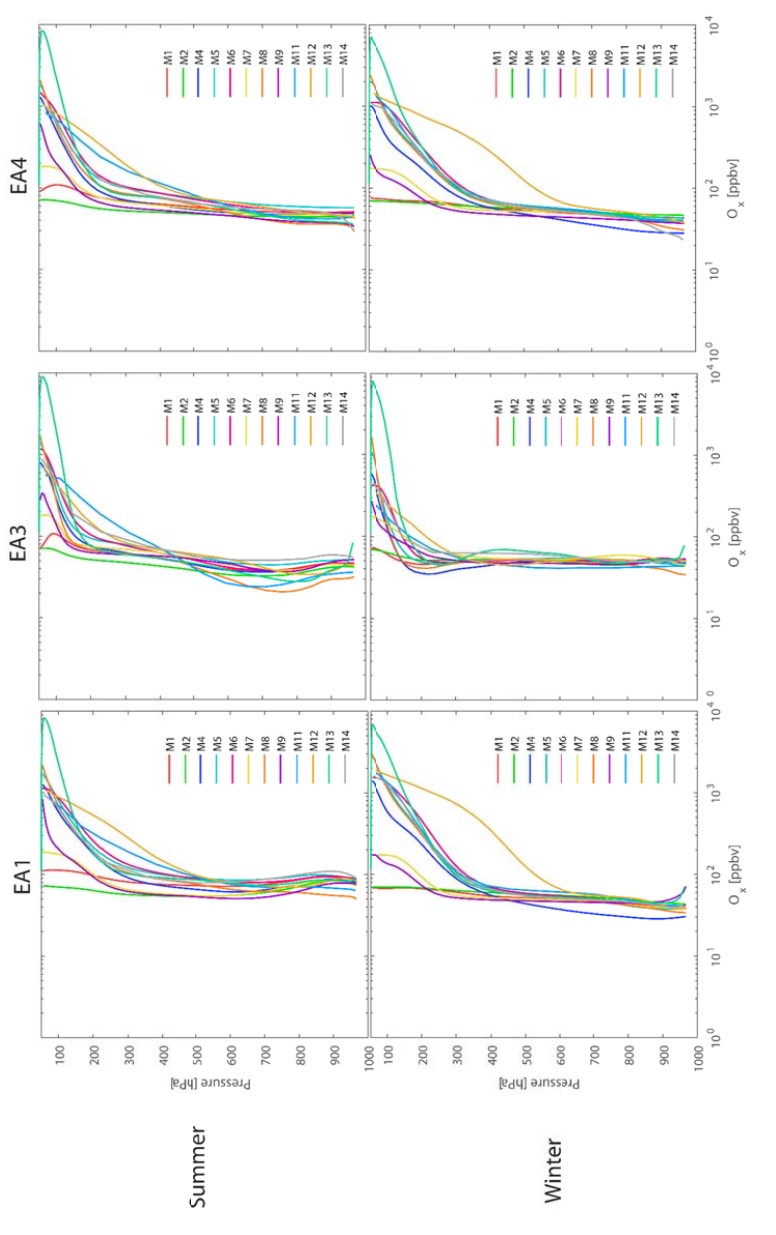

*Fig.9 Li et al., 2018*





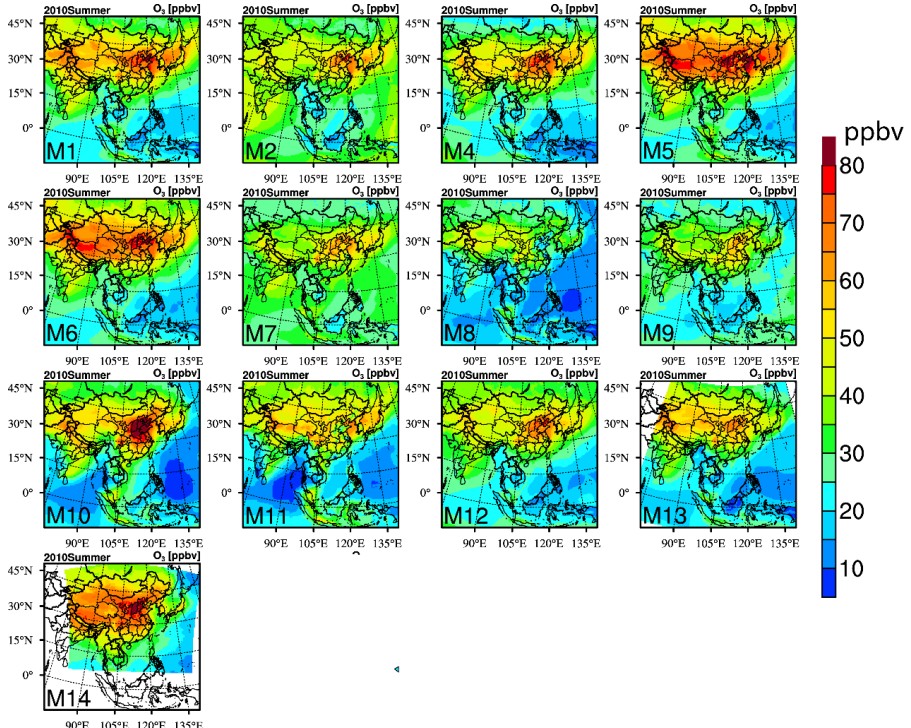

*Fig.10 Li et al.,2018*





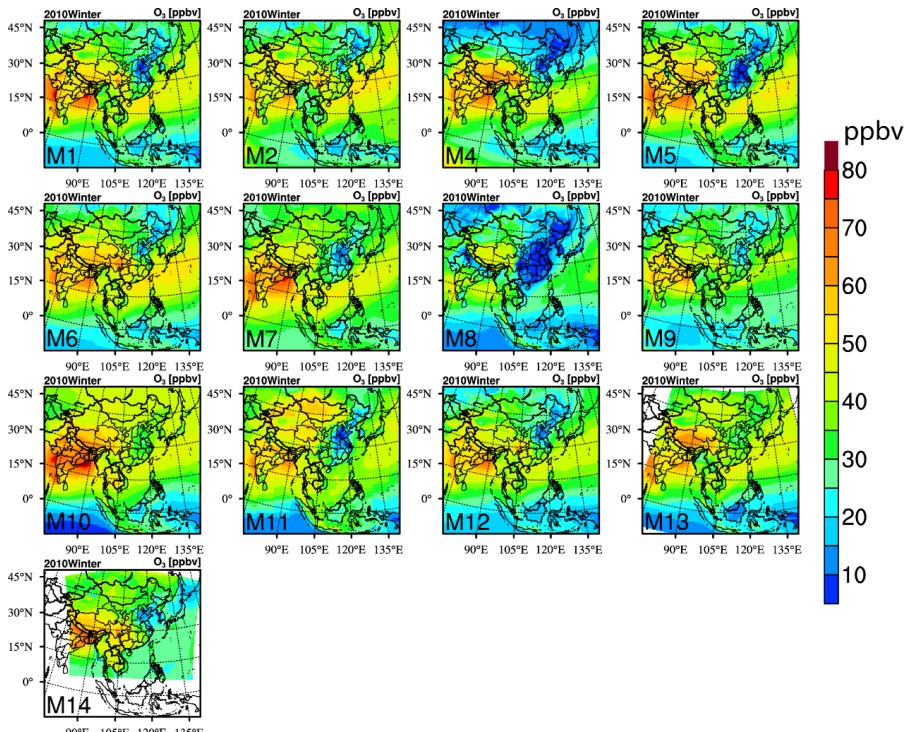

*Fig.11 Li et al.,2018*





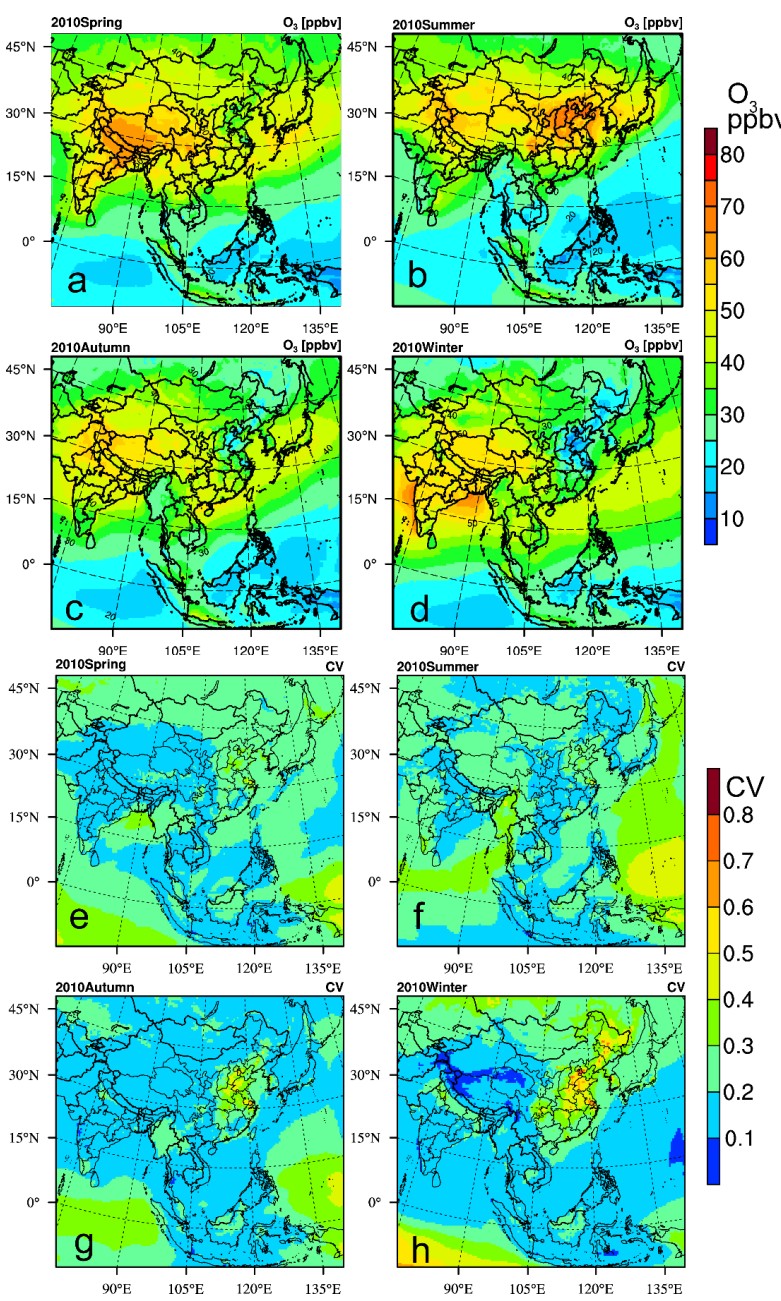

*Fig.12 Li et al.,2018*



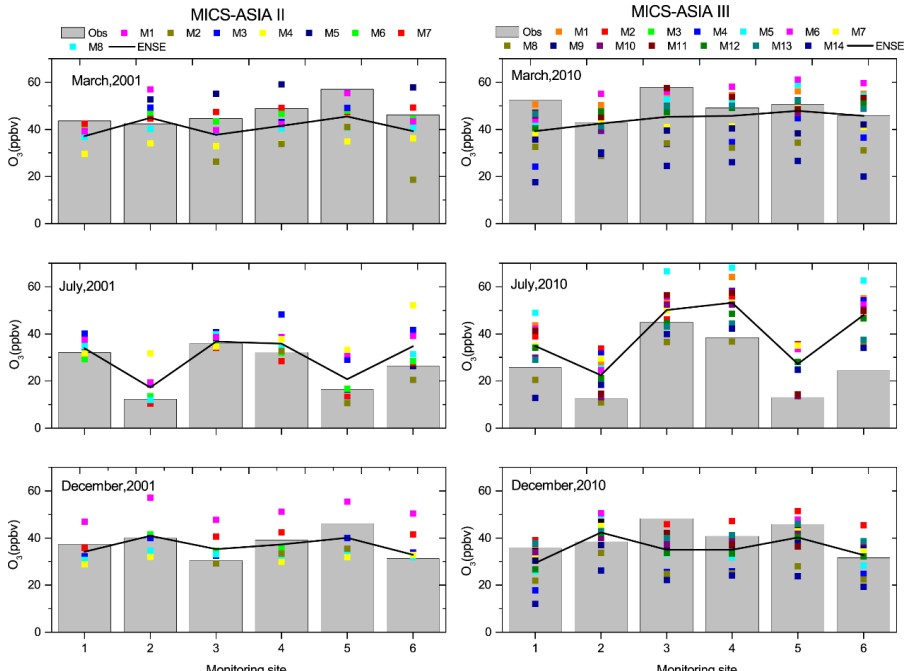

*Fig.13 Li et al., 2018*