# Peer review of "Model evaluation and inter-comparison of surface-level ozone and relevant species in East Asia in the context of MICS-Asia phase III Part I: overview"

_Atmospheric Chemistry and Physics, 2018_

## Referee Comment (RC1) · Anonymous Referee #1 · 16 Feb 2019

General comments:

This paper presents overview about Phase III of the chemical transport model inter-comparison sutudy MICS-ASIA for East Asia region. The atmospheric models participating in Phase III and its simulation framework have greatly improved from the previous MICS-ASIA Phase II. And, the calculation results are compared with the observations in industrial China, which was not done in the Phase II. So, this paper introducing MICS-ASIA Phased III is believed to have certain academic value. However, in the manuscript at the present time, there are many problems such as the sentences being too long, and the lack of the necessary information to convince the authors' in-

terpretation to the results. Then, the manuscript should be revised according to the following comments as well as many other specific comments before the publication in ACP.

1. About the length of the manuscript

It seems that the manuscript is too long compared to its contents. The things to be claimed should be focused (probably on what is stated in summary or the abstract), and the descriptions not related to those should be removed or simplified. The figures or their contents which are not necessary for the main line should be also omitted.

2. On the comparison of model results and measured values

Most models have rough resolution (horizontal direction: 45 km, vertical direction: 58 m near the ground), and it is not shown whether the observed values to be compared represent the extent of that range. If many measuring stations are unevenly distributed in a grid cell at locations with high NOx emissions, the effect of titration there is greater than the grid cell average. So, actually the models overestimating the measured ozone concentration may be correct.

3. About the investigation of intermodel variability on O3ãĂĂ(chapter.4)

In phase II of the MICS-ASIA, because input data (weather, emissions, boundary condition) are different, it was not possible to specify how much each process of chemistry, vertical diffusion, and dry deposition in the model contributed to calculated ozone variation among models. In the Phase III of this time, although common input data were provided to avoid it, it seems in this paper that the contribution of each of the above processes could not be specified again because the post process of these data differs between models. If the above guess is true, it seems better to clearly state it and to give up the brute forth evaluation of the contribution of each of the above process in sections 4.3-4.5. On the other hand, if you stick to say that you could specify the contribution of each of the above processes, you should add thoroughly the information

described in the following so that the reader can understand its rationality.

4. About authors' interpretation of the results

Many parts can not be convinced about the interpretation of the results by the author mainly because the differences among each model (e.g., differences of boundary conditions estimated with Mozart, Chaser, and by default settings, differences in dry deposition model, differences in sub-grid scale parameterization such as convection, differences in PBL model, and differences in spatiotemporal distribution of emissions) are not specifically mentioned. For relevant parts other than chapter 4, I will point out each of the following "other specific comments"".

Other specific comments:

p.5 L2-3

Is the problem (3) really addressed ? I don't think so, as I already mentioned in the general comments.

p.5 L10-11

You mean to interpolate model outputs to locations of observations both horizontally and vertically ? If yes, please show that method in detail. It may get rid of my concern mentioned in the general comments.

p.5 L24

Fig.1 does not introduce WRF model.

p.6 L28-p.7 L1

Please identify which model adopt the projection by themselves.

p.7 L5

I think two references should be moved after the names of the universities are introduced in L6.

p.7 L9

Are the models making boundary conditions depending on their own previous experience denoted by "default" in table 1 ? If yes, I think the phrase such as "their own" is better in table 1.

p.9 L4

Is the word "total" necessary ?

p.9 L5

M12 seems also an exception as well as M11.

p.9 L11-12

Is a two-peak seasonal cycle for O3 ? If yes , I see there are three peaks but not two. And I see observations show three-peak but not one-peak.

p.9 L22

"Similar results have been found in MICS-Asia II" seems contradict to the statement in L5-L7 of p.4.

p.10 L24-25

Show the evidence for the slight overestimation of 10 ppbv in M11 due to difficulties in dealing with vertical mixing.

p.10 L25-26

Show the evidence for the significant improvement of the model performance in winter, compared to in summer, due to the weak intensity of photochemical reactions.

p.11 L17

Add explanation how to derive the statics in table 2, 3 and 4 to clarify which part of the spatiotemporal deviations from the observations are included in the statics.

p.12 L12-13

Show the evidence for that the treatment of models on chemistry, vertical diffusion and dry deposition have contributed to the underestimation of NO.

p.13 L8-10

I can't understand why you selected the PBLH, emissions fluxes, dry deposition velocities, relationships between NOx and O3, amd the vertical profiles of O3 and its precursors to compare.

p.16 L23-L24

Jin et al (2015) perhaps showed the ozone formation regime at 1330 LST (overpass time of OMI) while you show that between 1000-1800 LST. Also, your results includes NOx titration effect while Jin et al (2015)'s results did not. So, I think it is not appropriate to compare them directly.,

p/17 L8-9

In M11, O3 does not seem positively correlated with NOx.

p.18 L17-18

Show the evidence that difference of concentrations are related to the treatments of convection and cloud activity among models.

p.19 L22-23

The locations of the place names shown in the text are not known for the foreign readers. So you should show these place names in Fig.10.

p.20 L16-17

Before you have the statement in L16-17, you should show that the wind fields are actually the same between the models which estimate 30 ppbv or higher O3 mixing ratio and those which estimate lower O3 mixing ratio. And, how do you think about the

difference of emissions that was discussed in section 4.2

p.33 L9

I guess the meteorological model used for providing meteorological fields with most models also use the domain in Fig.1. If yes, please mention about that too.

p.33 L14

Please add a description of the symbol such as "+" or "-" in Fig.2.

p.46 Fig.3 and p.47 Fig.4

The kinds of color of the curve in the figures is too many to distinguish. Are all the models need to be distinguished by different colors ?

Technical corrections:

p.3 L15

You need space between "2013" and "(Wang et al., 2017)". You can find the similar mistake to miss spaces elsewhere in the manuscript.

p.10 L4

"4)" should be removed.

p.19 L23

I think "predicated" should be "predicted".

p.20 L1

"EA1" should be moved right after "source regions"

---

## Referee Comment (RC2) · Anonymous Referee #2 · 20 Mar 2019

This paper describe the ability of an ensemble of regional chemistry-transport models to reproduce surface ozone pollution in East Asia as well as NOx concentrations. Indeed, recent observations do show that surface ozone concentrations are still increasing in China which underline the necessity to have good forecasting tools and means to set-up and control mitigation policies. This intercomparison is conducted in the framework of the Model Inter-Comparison Study for Asia phase III (MICS-ASIA III) which is the follow-up of MICS-ASIA II (2003) and MICS-ASIA I (1998). 13 models are cross compared for a one year simulation (2010). The simulation suits are based on state-of-the-art CTMs. Simulations are compared to available observations with especially observations available on industrialized China which was not the case of

[Figure]

MICS-ASIA II. Also the dispersion of the simulations are investigated to understand what reasons could explain models differences. Compared to European or American areas, the models have more difficulties to reproduced observed concentrations and the median of the ensemble do not always over skilled single models like it is the case for European ensembles. Such exercises have been proven useful to improve modelling suits and for this reason this paper is interesting for the community. The work conducted in that case is important and this study deserved to be published in ACP journal but corrections are probably needed to make the paper more efficient and to fulfill the high level standard of quality of the journal. I will list the comments and questions I still have on this work and that could help, i hope, to improve it. 1/ The analysis of the skills of an ensemble is always complicated. To be more clear and to have stronger messages, i suggest you to first analysis skills using the average of the ensemble and then to discuss the single models. By this way, it will allow to clearly identify the main biases either for seasonal analysis either for diurnal analysis and then discuss singularities . 2/ Maybe also it would nice to have a more explicit but still short reminder of the physical processes driving the variability in each sub-region (i.e late maxima of ozone in EA3 quite different than EA1 and even EA4). 3/ More informations about the nature of the stations and specifically about their representativity is needed. It is a key element of the model skills. Also for NO2 it exist sometimes biases (especially for stations far from sources) in the measurements when using molybden convertors devices since all nitrogen oxydes are measured instead of just NO2, do you have checked this ? 4/ I have the impression that authors do not need to include the EA2 region in the paper, you never use it in your discussions. 5/ Authors do evaluate several parameters relevant for model evaluation. It would have be better to have observations to put against models. It is often complicated to get all needed observations but maybe you can at list mention that in the prospectives. It become possible to have network ceilometers for PBLH evaluation. A lot of satellite observations are available to evaluate NOx or ozone at larger scales. What about vertical profiles ? Other comments etc . . . Page 3-Line 7 – Please remind the value of the threshold Page 10 - Line 4 – Please suppress

"4)" Page 10 – Line 18 – A good example where using the ensemble average allows to better structure the discussion and to be more precise on the model skills. Page 10 – Line 24-25 – "... due to difficulties in dealing with vertical mixing": how do we know that ? Page 12 – Line 16 – How statistics are calculated ? on hourly values ? Page 13 – Line 16 – Why choosing a sub selection of models ? It would be interesting to have all models. Page 14 – Line 3 – Von Engeln no ? Page 14 – Line 7 – You do not discuss VOC emissions. Would you suggest that models have no sensitivity to these emissions ? Page 14 – Line 15-20 – The discussion and the links between arguments are not that clear. Page 14 –Line 22 – I would say "net sink" since chemistry is a much higher absolute sink than deposition. Page 16 – Line 4 to 6 – Seems contradictory to have a small sink with considerable effect on oceanic surface. I would rather say that even if dry deposition velocities are small over oceanic surfaces, the impact of dry deposition over ocean is globally important because of the large surface ocean are representing. Page 16 – Line 6-8 – Why can we do the assumption that dry deposition is specifically important for EA4 ? Page 17 – Line 1 – I observe that range of concentrations for O3 and NOx can be very different between models but it is not clear if slopes are that different. Page 18 – Line 2 to 5 but also Line 7 to 20 – The variability authors are mentioning is not clear from figure 9. Also for differences between winter and summer, we need to have numbers to better evaluate this variability. Page 18 – Line 5-6 – Authors do have this information, it should more than an suggestion, no ? Page 19 – Line 8 – 9 – It is mention that dispersion between models is higher here than for the European case and authors suggest the models do not represent uncertainties, could you develop ? Also authors mention that key processes could miss, what kind of processes are they thinking to ? Page 20 – Line 11 to 15 – Do we observe same differences for higher levels ? Maybe in some models plumes are also present but at different altitudes. Page 21 – Line 2 – I'm not sure that author do define mathematically the coefficient of variation. Page 21 – Line 13 – Like in table1 authors do mention that "default" is used as boundary conditions. Default values should be more clearly defined ? climatology ? from where ? Page 22 –Line 7 – " .. its relevant species .." I also see VOC or even

radicals as relevant species for the tropospheric ozone cycle then it is better to mention 03 and NOx instead.

About Table and Figures Table2 – Maybe it is mandatory to mention how statistical indicator are calculated (i.e formula). Be careful "suqare" in the title instead of square. RMSE do have units, please mention it. Figure 1 – as mention earlier I would have removed EA2 that is not discussed. Figure 2 – probably too small as it is. The full black line does not seems necessary. Figure 9 – Maybe it is possible to reduce horizontal scale down to 10 ppb to have more space on the right and to better evaluate the ensemble dispersion. Figure 10 – Maybe too small also Figure 11 – Same as Figure 10
* * *

---

## Author Comment (AC1) · 30 Apr 2019

We thank Reviewer for his/her constructive comments.
Response to the Specific comments.

**General comments:** This paper presents overview about Phase III of the chemical transport model inter-comparison study MICS-ASIA for East Asia region. The atmospheric models participating in Phase III and its simulation framework have greatly improved from the previous MICS-ASIA Phase II. And, the calculation results are compared with the observations in industrial China, which was not done in the Phase II. So, this paper introducing MICS-ASIA Phased III is believed to have certain academic value. However, in the manuscript at the present time, there are many problems such as the sentences being too long, and the lack of the necessary information to convince the authors' interpretation to the results. Then, the manuscript should be revised according to the following comments as well as many other specific comments before the publication in ACP.

Reply: We totally agree with the reviewer. In the new manuscript, we accepted all comments suggested by the reviewer.

**Comment 1**:About the length of the manuscript. it seems that the manuscript is too long compared to its contents. The things to be claimed should be focused (probably on what is stated in summary or the abstract), and the descriptions not related to those should be removed or simplified. The figures or their contents which are not necessary for the main line should be also omitted.

Reply: We totally agree. In the revised manuscript, words have been cut back by 15-20%. 25% figures (Fig. 5. 6 and 11) and related discussions n (i.e. emissions) were also deleted. The revised manuscript included "1 Introduction; 2. Model validation(annual and monthly variation of surface $O_3$, NO and $NO_2$, surface $O_3$ diurnal variation, and $O_3$ vertical profiles); 3. Spatial distribution of $O_3$ and its comparison with MICS-Asia II, 4. Discussion (comparison with observed dry velocity and boundary layer height, relationships between $O_3$ with $NO_x$), 5. Summary"

**Comment** 2. On the comparison of model results and measured values. Most models have rough resolution (horizontal direction: 45 km, vertical direction: 58m near the ground), and it is not shown whether the observed values to be compared represent the extent of that range. If many measuring stations are unevenly distributed in a grid cell at locations with high NOx emissions, the effect of titration there is greater than the grid cell average. So, actually the models overestimating the measured ozone concentration may be correct.

Reply: We agree with the reviewer that the rough resolution may affect the model evaluation. In this study, observation data were taken from 1) Chinese Ecosystem Research Network (EA1); 2) Pearl River Delta Regional Air Quality Monitoring Network (PRD RAQMN) (EA2); 3) the Acid Deposition Monitoring Network in East Asia (EANET) (EA3). Observations were rarely affected by the very local emissions around sites, and were used to represent the regional air quality.

• As listed in Table R1 in this reply, most stations are located in rural, remote and clear urban regions in EA1. Fig. R1 presents the scatter plots of NO emissions in 45 and 3km emission inventory. Emission errors resulting from coarse grids were not significant in most stations. This implied that observation generally represents the 45km averages of ozone.

Table R1site descriptions in Chinese Ecosystem Research Network

| Site | Site characteristics | Longitude, latitude | |
|------|---------------------|---------|--------|
| Xinglong | Remote | 117.576 | 40.394 |
| Lingshan | Remote | 115.431 | 39.968 |
| Yangfang | Rural | 116.11 | 40.13 |
| Xianghe | Suburban | 116.962 | 39.754 |
| Langfang | Suburban | 116.689 | 39.549 |
| Zhuozhou | Suburban | 115.99 | 39.46 |
| Datong | Suburban | 113.389 | 40.089 |
| Zhangjiakou | Suburban | 114.918 | 40.771 |
| Cangzhou | Suburban | 116.779 | 38.286 |
| Yanjiao | Suburban | 116.824 | 39.961 |
| Beijing | Urban | 116.372 | 39.974 |
| Baoding | Urban | 115.441 | 38.824 |
| Shijiazhuang | Urban | 114.529 | 38.028 |
| Chengde[*] | Urban | 117.925 | 40.973 |
| Tianjin | Urban | 117.206 | 39.075 |
| Tanggu[*] | Urban | 117.717 | 39.044 |
| Caofeidian[*] | Urban | 118.442 | 39.270 |
| Tangshan | Urban | 118.156 | 39.624 |
| Qian'an[*] | Urban | 114.800 | 40.100 |

*cities are clear, and annual $PM_{2.5} < 35$ μg/m$^3$

[Figure]

Fig.R1 Scatter plots of NO emission rates (μg/m$^2$/s) at observation sites in EA1 in 45km and 3km resolution emission inventory (MEIC)

• Pearl River Delta Regional Air Quality Monitoring Network (PRD RAQMN) was jointly established by the Guangdong Provincial Environmental Monitoring Centre (GDEMC) and the Environmental Protection Department of the Hong Kong Special Administrative Region (HKEPD) from 2003 to 2005. The PRD RAQMN was to probe the regional air quality, assess the effectiveness of emission reduction measures and enhance the roles of monitoring networks in characterizing regional air quality and supporting air quality management (Zhong et al.,2013). So sites are rarely affected by the local emissions near them. Fig. R2 showed the Spatial distribution of average concentrations of $NO_2$ and $O_3$ in the PRD-RAQMN Network. Concentrations of pollutants were smooth. The effect of very local emissions was rarely seen.

[Figure]

Fig.R2 Spatial distribution of average concentrations of $NO_2$ and $O_3$ in the PRD-RAQMN Network, figure is annual report of Pearl River Delta Regional Air Quality Monitoring Network in 2013 (https://www.epd.gov.hk/epd/sites/default/files//epd/english/resources_pub/publications/files/PRD_2013_report_en.pdf)

• Sites in EANET are mostly located in oceanic regions (Hedo, Ogasawara and Oki) and remote regions (Rishiri, Ochiishi, Yusuhara, Sado-seki, Happo). More information can be found in Ban et al. (2016).

**Comment** 3. About the investigation of intermodel variability on O3ãˇAˇA(chapter.4) In phase II of the MICS-ASIA, because input data (weather, emissions, boundary condition) are different, it was not possible to specify how much each process of chemistry, vertical diffusion, and dry deposition in the model contributed to calculated ozone variation among models. In the Phase III of this time, although common input data were provided to avoid it, it seems in this paper that the contribution of each of the above processes could not be specified again because the post process of these data differs between models. If the above guess is true, it seems better to clearly state it and to give up the brute forth evaluation of the contribution of each of the above process inspections 4.3-4.5. On the other hand, if you stick to say that you could specify the contribution of each of the above processes, you should add thoroughly the information described in the following so that the reader can understand its rationality.

Reply: We totally agree. In MICS-Asia III, we found that there were significant model biases and intermodel variability in summer ozone in North China Plain and Western Pacific. These findings were not revealed in phase II of MICS-Asia. This point is beyond we expected before MICS-Asia III. Hence, one issue we are facing is to explain the bias causes or provide a future direction on analysis for MICS-Asia IV. We agree the reviewer that quantifying the contribution of each process processes (vertical mixing, horizontal advection, gaseous and heterogeneous chemistry, dry and wet deposition, emissions and model resolution…) is important to explain model bias. Sensitivity simulation is a good way. But this requires a tremendous amount of computational cost and data space for 14 models. Designing sensitivity simulating scenarios with acceptable costs is essential to next studies. The MICS-Asia III has not directly output the contribution of each process, so we did a qualitative analysis on potential causes by comparison between models and observations to narrow sensitivity simulating scenarios for MICS-Asia IV. We believe that this is also helpful for other model developers to improve model performance in East Asia. In MICS-Asia II, related discusses were mostly based on guesses because meteorology, emissions, model domain, boundary conditions were quite different. In MCIS-Asia III, common input data provide a good chance for this qualitative analysis.

We agree with the reviewer that brute forth evaluation of the contribution of processes may cause errors or uncertainties. In the revised manuscript, we collected observation data on key parameters of potential processes as much as possible. Our focus was the model evaluation on these parameters, which has not been conducted by previous phase of MICS-Asia. So we changed the title from "Investigation of intermodel variability on $O_3$" to "Discussion".

As shown in Fig. R3, ensemble average dry deposition velocity of $O_3$ underestimated observations in August-September by 30-50% in EA1. This underestimation decreased the deposition amounts of surface $O_3$ and partly explained the overestimation of ensemble simulated $O_3$ in summer. This is consistent with intermodel comparison between M11 with M1-M6. M11 reproduced observed surface $O_3$ in EA1in May-July. The higher dry deposition velocities in M11 between May-July (0.3 cm/s) contributed to low surface $O_3$ than M1-M6. This implied that we should conducted the sensitivity analysis on dry deposition to quantify its impact on EA1 surface ozone in MICS-Asia IV. In EA4, simulated dry deposition velocity agreed well with observations, so there could be other reasons responsible for overestimation in EA4.

Previous studies revealed that $O_3$ precursors are mostly constrained within the boundary layer (Quan et al., 2013). The model evaluation on PBLH and turbulent kinetic energy is essential for the interpretation of model biases with observations. Unfortunately, few observations on turbulent kinetic energy were directly measured in East Asia. Fig. R4 presents the comparison between simulated and observed PBLH. In EA1, all the selected models exhibited the spring-maximum and winter-minimum season cycle, which captured the major pattern of climatology of PBLH observations (Guo et al.,2016). The Ense on PBLH was 100-200 m higher than radiosonde

measurements. This is likely caused by the inconsistency of samples between models and measurements. The simulation was the mean value of 12 hours (08:00-20:00), while the average of measurements was calculated based on 3 hours (08:00, 14:00 and 20:00). In MICS-Asia IV, more model evaluation on turbulent kinetic energy is urgent.

[Figure]

Fig. R3 Simulated and observed monthly dry deposition velocity

In the revised manuscript, we moved vertical profile of $O_3$ into the section "model evaluation", and observations in EA3 and EA4 were added. In general, ensemble means (Ense) presented an underestimation and overestimation for EA3 $O_3$ in middle (500-800 hpa) and lower (below 900 hpa) troposphere, respectively. In winter, the underestimation even extended to 200hpa in winter. The magnitudes of underestimation and overestimation reached 10-40 ppbv and 10-20 ppbv. In EA4, Ense reproduced the vertical structure of ozone in both summer and winter. An overestimation existed below 800 hpa in summer, with a magnitude of 10-20 ppbv.

[Figure]

now.

Fig. R4 Simulated and observed monthly daytime PBLH

[Figure]

Fig. R5 Simulated and observed O₃ profiles in summer and winter of 2010, averaged over all observed stations in three subregions over East Asia (EA1: left column, EA3: middle column, EA4: bottom column).

The evaluation on chemistry in models is a difficult problem all along. As far as

we know, there are no direct measurement on ozone production rates in East China till now. The relationships between $O_3$ with its precursors usually was regarded as an effective index on chemistry. We realized that the simple comparison between $O_3$ with $NO_x$ could bring errors or uncertainties. Hence, the relationship only was used to qualitative analyze the intermodel variability on chemistry, more quantitative analysis will be conducted in MICS-Asia IV. We believe that this qualitative analysis is helpful to model developer. For example, we found that the slope and intercept between $O_3$ and $NO_x$ in M11 (the best performance of $O_3$ in EA1) were closer to observations. The lower slope (-1.02) in M11 than M1-M6 (-1.31 - -2.25) indicated a weaker ozone chemical production intensity. This is validated by Akimoto et al. (2019) in which ozone chemical production in M11 was 60% of M1.

[Figure]

Fig. R9 Scatter plots between monthly daytime (08:00-20:00) surface $NO_x$ and $O_3$ at each station over EA1(red), EA3(green)and EA4(blue) in May-October, for observations(obs) and models

**Comment** 4. About authors' interpretation of the results. Many parts cannot be convinced about the interpretation of the results by the author mainly because the differences among each model (e.g., differences of boundary conditions estimated with Mozart, Chaser, and by default settings, differences in dry deposition model, differences in sub-grid scale parameterization such as convection, differences in PBL model, and differences in spatiotemporal distribution of emissions) are not specifically mentioned. For relevant parts other than chapter 4, I will point out each of the following "other specific comments"

Reply: We understand the reviewer. The large divergence on parameterizations and

emissions among models is always a difficult problem in air quality model intercomparison projects. Hence, some intercomparison projects like HTAP v1 conducted by United Nations, CityDelta by Europe Union and AQMEII employed models with different resolutions and various meteorology. Sometimes, different lateral boundary conditions were used in regional models (CityDelta, AQMEII). This increased the difficulty of interpretation. In MICS-Asia III, most models employed the same emissions, meteorology and resolution, which provide a good chance to explore the impact of parameterization on ozone.

As mentioned by the reviewer, no specifying the contribution of processes could bring errors or uncertainties to the interpretation of the results. So we moved our focus from interpretation of the results to the model evaluation on key parameters of processes by collecting their observations (dry deposition velocity, PBLH, vertical profiles) as much as possible. We hope our analysis is helpful to detailed model intercomparison in next studies and other model developers in East Asia.

We revised our manuscript according to your flowing comments.

**"Other specific comments:**
**Comment 5:** p.5 L2-3 Is the problem (3) really addressed? I don't think so, as I already mentioned in the general comments

Reply: In the revised manuscript, we deleted the problem3.

**Comment 6:** p.5 L10-11You mean to interpolate model outputs to locations of observations both horizontally and vertically? If yes, please show that method in detail. It may get rid of my concern mentioned in the general comments.

Reply: Firstly, we determine the model grid cell indexes of observation sites from their longitude, latitude, and height above sea levels. If there are two or more sites in one grid, we will select their mean values to compare with model outputs in this grid.

In the revised manuscript, we added related descriptions.

**Comment 7:** p.5 L24 Fig.1 does not introduce WRF model.

Reply: In the revised manuscript, we added a description "The domain of meteorological fields is shown in Fig.1".

**Comment 8:** p.6 L28-p.7L1 Please identify which model adopt the projection by themselves.

Reply: M13 and M14 made the projection by themselves

**Comment 9:** p.7 L5 I think two references should be moved after the names of the universities are introduced in L6

Reply: We revised it..

**Comment 10:** p.7 L9 Are the models making boundary conditions depending on their

own previous experience denoted by "default" in table 1? If yes, I think the phrase such as "their own" is better in table 1.

Reply: We revised it.

**Comment 11:** p.9 L4 Is the word "total" necessary?

Reply: We deleted it.

**Comment 12:** p.9 L5 M12 seems also an exception as well as M11.

Reply: We agree, and revised it in the new manuscript.

**Comment 13:** p.9 L11-12 Is a two-peak seasonal cycle for O3? If yes, I see there are three peaks but not two. And I see observations show three-peak but not one-peak.

Reply: We revised this sentence. "In EA3, most models (except M7, M8 and M11) exhibited high $O_3$ concentrations in March-May and September-November. Observed O3 showed that the highest concentrations appeared in October-November."

**Comment 14:** p.9 L22"Similar results have been found in MICS-Asia II" seems contradict to the statement in L5-L7 of p.4.

Reply: Thanks. In L5- L7 of P4, the underestimation of simulated O3 appeared in spring (March) and winter (December) during the MCS-Asia II. In this study, our reported overestimation of $O_3$ was in May-October (L22 P9). The periods in P4 and P9 are different.

**Comment 15:** p.10 L24-25 Show the evidence for the slight overestimation of 10 ppbv in M11 due to difficulties in dealing with vertical mixing.

Reply: In M11, the minimum of vertical diffusivity was set to be 0.5 $m^2$ $s^{-1}$. This value is a little higher than other models (e.g. CAMx: 0.1 $m^2$ $s^{-1}$). In the stable boundary layer on nighttime, the higher vertical diffusivity may transport high ozone in upper layer to the surface, and also uplifted surface NO. The lower NO weakens the ozone titration.

We realized that vertical mixing may be not the only reason of nighttime ozone overestimation in M11. We needed more observed evidence to support our conclusion. So, we deleted it in the revised manuscript.

**Comment 16:** p.10 L25-26 Show the evidence for the significant improvement of the model performance in winter, compared to in summer, due to the weak intensity of photochemical reactions.

Reply: Thanks. As shown in Table R2, ensemble simulated ozone (Ense) in winter was closer to observations than summer. The ratio between Ense and Observation was 1.28, much lower than 1.69 in summer. The intensity of overestimation increased from winter

to summer, with the increase of solar radiation. This implied that the treatment of photochemical reactions in models may play an important role in this overestimation.

Table R2 Observed and ensemble simulated ozone (Ense) in EA1

| Season | Observation | Ense | Ense/Obs |
|---|---|---|---|
| Winter (Dec-Feb) | 12.6 | 16.1 | 1.28 |
| Spring (Mar-May) | 25.6 | 34.6 | 1.35 |
| Summer (Jun-Aug) | 38.0 | 64.4 | 1.69 |
| Autumn (Sep-Nov) | 14.9 | 23.6 | 1.58 |

**Comment 17:** p.11 L17Add explanation how to derive the statics in table 2, 3 and 4 to clarify which part of the spatiotemporal deviations from the observations are included in the static

Reply: We add the definition of these statics in Appendix A in the revised manuscript.

**Comment 18:** p.12 L12-13 Show the evidence for that the treatment of models on chemistry, vertical diffusion and dry deposition have contributed to the underestimation of NO.

Reply: Thanks. We delete this sentence.

**Comment 19:** p.13 L8-10 I can't understand why you selected the PBLH, emissions fluxes, dry deposition velocities, relationships between NOx and O3, and the vertical profiles of O3 and its precursors to compare.

Reply: Thanks for your comments. In the revised manuscript, we collected related observations to evaluate the model performance, as discussed in Comment 3.

**Comment 20:** p.16 L23-L24Jin et al (2015) perhaps showed the ozone formation regime at 1330 LST (overpasstime of OMI) while you show that between 1000-1800 LST. Also, your results include NOx titration effect while Jin et al (2015)'s results did not. So, I think it is not appropriate to compare them directly.,

Reply: We agree. In the revised manuscript, we deleted this reference.

**Comment 21:** p/17 L8-9 In M11, O3 does not seem positively correlated with NOx.

Reply: Sorry. M9 and M10 were positively correlated with NOx, instead of M8 and M11. In the revised manuscript, we revised it.

**Comment 22:** p.18 L17-18 Show the evidence that difference of concentrations are related to the treatments of convection and cloud activity among models.

Reply: Thanks. Fig. R5 showed the simulated and observed $O_3$ profiles in EA3. Clearly,

the most significant underestimation and inter-variability of models appeared in 950-700 hpa (~0.5-2.5 km). The climatology of ozone sounding revealed a high relative humidity (about 80%) and enhanced ozone layer in this layer (0.5-2 km) in summer (Leung et al., 2003). Leung et al. (2004) stated that the ozone in this layer was likely from convection of photochemical production in the polluted boundary layer, based on the simultaneous occurrence of high ozone mixing ratio and high relative humidity. In MICS-Asia III, horizontal resolution is 45 km, which was not enough to explicitly simulate the convection. So sub-grid parameterization in models may played an important in the underestimation and inter-variability. We realized that these are not direct evidence because impact of convections in models were not output. Hence, we delete this sentence in the revised manuscript.

[Figure]

Fig. R10 Seasonally averaged ozone profiles in the troposphere above Hong Kong summer

**Comment 23:** p.19 L22-23 The locations of the place names shown in the text are not known for the foreign readers. So, you should show these place names in Fig.10.

Reply: We plotted place names in Fig. R11 in the revised manuscript.

[Figure]

① Bohai Sea
② East China Sea
③ Korea
④ Japan
⑤ The Sea of Japan

Fig. R11 Locations of related regions

**Comment 24:** p.20 L16-17 Before you have the statement in L16-17, you should show that the wind fields are actually the same between the models which estimate 30 ppbv or higher O3 mixing ratio and those which estimate lower O₃ mixing ratio. And, how do you think about the difference of emissions that was discussed in section 4.2

Reply: We agree. In the revised manuscript, we showed the simulated wind fields by models. Winds between models were similar. In section 4.2, we found that EA1 emissions in M1, M4 and M11 are similar, but the simulated O₃ between these three models the western Pacific Ocean showed a O₃ discrepancy. So, there could be other causes responsible for this discrepancy, besides emissions in source regions.

[Figure]

Fig. simulated surface wind velocities(m/s) in MICS-Asia III

**Comment 25** p.33 L9 I guess the meteorological model used for providing meteorological fields with most models also use the domain in Fig.1. If yes, please mention about that too.

Reply: We added this point in the revised manuscript.

**Comment 26** p.33 L14 Please add a description of the symbol such as "+" or "-" in Fig.2.

Reply: We added a description in the caption of Fig.2.

**Comment 27:** p.46 Fig.3 and p.47 Fig.4 The kinds of color of the curve in the figures is too many to distinguish. Are all the models need to be distinguished by different colors?

Reply: Sorry for trouble you in Fig.2 and 3. An aim of MICS-Asia III is to examine the models' performance for O₃ in East Asia, and provide useful information to improve model ability. As the first step, we need discuss the strengths of individual models and tell the readers as much as possible. Then we will compare the parametrization of this model with others and explore why it exhibit a better performance. In this respect we need label each model in Fig.2 and 3. We listed the performance of individual models in section 3.2. For example, we mentioned that M11 was closer to O₃ observations in EA1. In our another manuscript, we compared M11 parametrization of transport, vertical diffusion and heterogeneous chemistry with M1 and M6. This is helpful to improve the model.

**Technical corrections:**

**Comment 28:** p.3 L15 You need space between "2013" and "(Wang et al., 2017)". You can find the similar mistake to miss spaces elsewhere in the manuscript.

Reply: We revised it

**Comment 29:** p.10 L4"4)" should be removed.

Reply: We revised it

**Comment 30:** p.19 L23 I think "predicated" should be "predicted".

Reply: We revised it

**Comment 31:** p.20 L1"EA1" should be moved right after "source regions

Reply: We revised it

References:

Akimoto, H., Nagashima, T., Li, J., Fu, J. S., Ji, D., Tan, J., and Wang, Z.: Comparison of surface ozone simulation among selected regional models in MICS-Asia III – effects of chemistry and vertical transport for the causes of difference, Atmos. Chem. Phys., 19, 603-615, https://doi.org/10.5194/acp-19-603-2019, 2019.

Ban, S. , Matsuda, K. , Sato, K. , & Ohizumi, T. . (2016). Long-term assessment of nitrogen deposition at remote EANET sites in japan. Atmospheric Environment, 146, 70-78.

Guo, J., Miao, Y., Zhang, Y., Liu, H., Li, Z., Zhang, W., He, J., Lou, M., Yan, Y., Bian, L., and Zhai, P.: The climatology of planetary boundary layer height in China derived from radiosonde and reanalysis data, *Atmos. Chem. Phys*., 16, 13309-13319, https://doi.org/10.5194/acp-16-13309-2016, 2016

Han, Z., Sakurai, T., Ueda, H., Carmichael, G. R., Streets, D., Hayami, H., Wang, Z., Holloway, T., Engardtg, M., Hozumib, Y., Parkh, S.U., Kajinoi, M., Sarteletj, K., Fungk, C., Bennetg, C., Thongboonchooc, N., Tangc, Y., Changk, A., Matsudal, K., Amannm, M. : MICS-ASIA II: model intercomparison and evaluation of ozone and relevant species, Atmos. Environ., 42(15), 3491-3509,2008.

Leung Y K , Chang W L , Chan Y W . Some characteristics of ozone profiles above Hong Kong. Meteorology and Atmospheric Physics, 87(4):279-291, 2004.

Liuju Zhong, Peter K.K. Louie*, Junyu Zheng, K.M. Wai, Josephine W.K. Ho, Zibing Yuan, Alexis K.H. Lau, Dingli Yue, Yan Zhou, The Pearl River Delta Regional Air Quality Monitoring Network – Regional Collaborative Efforts on Joint Air Quality Management, Aerosol and Air Quality Research, 13: 1582–1597, 2013

Quan, J., Tie, X., Zhang, Q., Liu, Q., Li, X., Gao, Y., and Zhao, D.: Evolution of planetary boundary layer under different weather conditions, and its impact on aerosol

concentrations, Particuology, 11(1), 34-40, 2013.

Zhong, L., Louie, P., Zheng, J., Wai, K. M., Josephine W.K. Ho, Yuan, Z., Lau, A. K. H., Yue, D., Zhou, Y.: The Pearl River Delta Regional Air Quality Monitoring Network – Regional Collaborative Efforts on Joint Air Quality Management, Aerosol and Air Quality Research, 13: 1582–1597, 2013

---

## Author Comment (AC2) · 30 Apr 2019

We thank Reviewer for his/her constructive comments.
Response to the Specific comments.

**General comments:** This paper describe the ability of an ensemble of regional chemistry-transport models to reproduce surface ozone pollution in East Asia as well as NOx concentrations. Indeed, recent observations do show that surface ozone concentrations are still in-creasing in China which underline the necessity to have good forecasting tools and means to set-up and control mitigation policies. This intercomparison is conducted in the framework of the Model Inter-Comparison Study for Asia phase III (MICS-ASIA III) which is the follow-up of MICS-ASIA II (2003) and MICS-ASIA I (1998). 13 models are cross compared for a one-year simulation (2010). The simulation suits are based on state-of-the-art CTMs. Simulations are compared to available observations with specially observations available on industrialized China which was not the case of MICS-ASIA II. Also, the dispersion of the simulations are investigated to understand what reasons could explain models differences. Compared to European or American are as, the models have more difficulties to reproduced observed concentrations and the median of the ensemble do not always over skilled single models like it is the case for European ensembles. Such exercises have been proven useful to improve modelling suits and for this reason this paper is interesting for the community. The work conducted in that case is important and this study deserved to be published in ACP journal but corrections are probably needed to make the paper more efficient and to fulfill the high level standard of quality of the journal. I will list the comments and questions I still have on this work and that could help, i hope, to improve it.

Reply: Thanks a lot for your insightful comments. We accept all your comments in the revised manuscript.

**Comment 1:** The analysis of the skills of an ensemble is always complicated. To be more clear and to have stronger messages, i suggest you to first analysis skills using the average of the ensemble and then to discuss the single models. By this way, it will allow to clearly identify the main biases either for seasonal analysis either for diurnal analysis and then discuss singularities.

Reply: We totally agree. We firstly evaluate the ensemble performance in each section of the revised manuscript.

In section 3.1,

"The $O_3$ NMB and RMSE of ensemble mean were significantly less than the ensemble median in most situations (Table 1). Therefore, we only presented the results of multi-model mean ensemble (Ense). In general, the majority of models significantly overestimated annual surface $O_3$ compared with the observations in EA1, EA3 and EA4 (Fig. 2). Ense overestimated surface $O_3$ by 10-15 ppbv in these subregions. Ense $NO_2$ was generally close to the observations to within ±20% in all subregions. In EA1 and EA3, Ense NO was 5-10 ppbv lower than observation, and showed a reasonable

performance in EA4."

In section 3.2,

"From the perspective of monthly variation, the overestimation of $O_3$ mostly appeared in May-September in EA1. Ense $O_3$ was 10-30 ppbv higher than observations, 30-70% of observed values. In the same period (May-September), Ense NO and $NO_2$ appeared to be consistent with observations, attaining mean biases of < 3 ppbv. This suggests that the intercomparison on $O_3$ production efficiency per $NO_x$ with observations is needed. In EA3, Ense $O_3$ agreed well with observed high autumn $O_3$, but overestimated from January to September by 5-15 ppbv (15-60% of observations). This maximum of overestimation appeared in March-April (15ppbv), which led to a spring peak in simulated $O_3$ which was not found in observations. This overestimation was partly related to the underestimation of NO in the same months, which decreased the titration effect. For $NO_2$, Ense agreed well with observed values in June-December, and slightly underestimated observations in January-May. In EA4, a significant overestimation of $O_3$ and underestimation of NO existed in June-October. Both observations and Ense NO were lower than 0.5 ppbv, so impact of by NO underestimation on $O_3$ are needed to be further explored. The ensemble $NO_2$ was generally close to the observations to within ±0.5 ppbv."

In section 3.3,

"In general, model results for three sub-regions exhibited a larger spread with a magnitude of 10-50 ppbv throughout the diurnal cycle than that in Europe and North America (Solazzo et al., 2012). The Ense $O_3$ in summer exhibited a systematic overestimation (20 ppbv) throughout the diurnal cycle in EA1. This indicated that models had difficulty dealing with $O_3$ in North China Plain. Compared with summer, there was only a slight systematic overestimation of Ense $O_3$ in other seasons (3-5 ppbv). In EA3, Ense $O_3$ generally agreed with the observations in summer, autumn and winter. In particular, the $O_3$ maximum around noon was reproduced, reasonably. There was only a 3-5 ppbv overestimation during 16:00-23:00 and early morning (6:00-10:00). In spring, a systematic overestimation of Ense $O_3$ exited in the whole diurnal cycle (5-10 ppbv). In EA4, Ense captured the small diurnal variation of $O_3$ in four seasons, but significantly overestimated observations in summer and autumn (5-20 ppbv). In spring and winter, differences between Ense and observations were within 5 ppbv."

In section 3.4,

"In general, Ense performed a better performance level than individual models for representing $NO_2$ in East Asia, reproducing the observed seasonal cycle and magnitudes. However, Ense did not always exhibited a superior performance for $O_3$ over certain individual model in East Asia, which was in contrast to its performance in Europe . M11 and M7 agreed well with observations in EA1 and EA3, while ENSE tended to overestimate $O_3$ concentrations in May-September in EA1 and January-September in EA3. Loon et al. (2007) indicated that ENSE exhibited a superior performance level

only when the spread of ensemble-model values was representative of the uncertainty of $O_3$. This indicated that most models did not reflect this uncertainty or missed key processes in MICS-Asia III."

In section 3.5,

"In general, ensemble means (Ense) presented an underestimation and overestimation for EA3 $O_3$ in middle (500-800 hpa) and lower (below 900 hpa) troposphere, respectively. In winter, the underestimation even extended to 200hpa in winter. The magnitudes of underestimation and overestimation reached 10-40 ppbv and 10-20 ppbv. In EA4, Ense reproduced the vertical structure of ozone in both summer and winter. An overestimation existed below 800 hpa, with a magnitude of 10-20 ppbv."

**Comment 2**:Maybe also it would nice to have a more explicit but still short reminder of the physical processes driving the variability in each sub-region (i.e late maxima of ozone in EA3 quite different than EA1 and even EA4).

Reply: We totally agree. In the revised manuscript, we discussed the physical factors driving variability of each region on seasonal cycle.

"The East Asia monsoon played an important role in seasonal cycle of $O_3$ in subregions by the long-range transport. Besides local intensive photochemical productions, the $O_3$ summer maxima in EA1were also affected by regional transport from Yangtze River Delta under prevailed summer southern monsoon (~20%) (Li et al., 2016). In EA3, a late maximum of $O_3$ in September-November was quite different from EA1 and EA4. This is largely attributed to the long-range transport of $O_3$ and its precursors in the polluted continental air masses from northern China and photochemical formation under dry and sunny weather conditions in autumn (Zheng et al., 2010). In EA4, the seasonal change of $O_3$ concentrations was characterized by two peaks in spring and autumn. The first and second peak in March–April and May-June were mainly influenced by the inflow from outside of East Asia and chemically produced $O_3$ by regional emissions, respectively. In the next studies, we will conduct the intermodel comparison on transport fluxes of $O_3$ between sub-regions over East Asia."

**Comment 3**:More informations about the nature of the stations and specifically about their representativity is needed. It is a key element of the model skills. Also, for NO2 it exist sometimes biases (especially for stations far from sources) in the measurements when using molybden convertors devices since all nitrogen oxydes are measured instead of just NO2, do you have checked this?

Reply: We agree. In this study, stations are taken from from 1) Chinese Ecosystem Research Network (EA1); 2) Pearl River Delta Regional Air Quality Monitoring Network (PRD RAQMN) (EA2); 3) the Acid Deposition Monitoring Network in East Asia (EANET) (EA3). Observations were rarely affected by the very local emissions around sites, and were used to represent the regional air quality.

- As listed in Table R1 in this reply, most stations are located in rural, remote and

clear urban regions in EA1. Fig. R1 presents the scatter plots of NO emissions in 45 and 3km model grid cell. Clearly, emission errors resulting from coarse grids were not significant in most stations. This implied that observation generally represents the 45km averages of ozone.

Table R1site descriptions in Chinese Ecosystem Research Network

| Site | Site characteristics | Longitude, latitude | |
|---|---|---|---|
| Xinglong | Remote | 117.576 | 40.394 |
| Lingshan | Remote | 115.431 | 39.968 |
| Yangfang | Rural | 116.11 | 40.13 |
| Xianghe | Suburban | 116.962 | 39.754 |
| Langfang | Suburban | 116.689 | 39.549 |
| Zhuozhou | Suburban | 115.99 | 39.46 |
| Datong | Suburban | 113.389 | 40.089 |
| Zhangjiakou | Suburban | 114.918 | 40.771 |
| Cangzhou | Suburban | 116.779 | 38.286 |
| Yanjiao | Suburban | 116.824 | 39.961 |
| Beijing | Urban | 116.372 | 39.974 |
| Baoding | Urban | 115.441 | 38.824 |
| Shijiazhuan | Urban | 114.529 | 38.028 |
| Chengde[*] | Urban | 117.925 | 40.973 |
| Tianjin | Urban | 117.206 | 39.075 |
| Tanggu[*] | Urban | 117.717 | 39.044 |
| Caofeidian[*] | Urban | 118.442 | 39.270 |
| Tangshan | Urban | 118.156 | 39.624 |
| Qian'an[*] | Urban | 114.800 | 40.100 |

*cities are clear, and annual $PM_{2.5}<35$ μg/m$^3$

[Figure]

Fig.R1 Scatter plots of NO emission rates (μg/m$^2$/s) at observation sites in EA1 in 45km and 3km resolution emission inventory

- Pearl River Delta Regional Air Quality Monitoring Network (PRD RAQMN) was

jointly established by the Guangdong Provincial Environmental Monitoring Centre (GDEMC) and the Environmental Protection Department of the Hong Kong Special Administrative Region (HKEPD) from 2003 to 2005. The PRD RAQMN was to probe the regional air quality, assess the effectiveness of emission reduction measures and enhance the roles of monitoring networks in characterizing regional air quality and supporting air quality management (Zhong et al.,2013). So sites are rarely affected by the local emissions near them. Fig. R2 showed the Spatial distribution of average concentrations of $NO_2$ and $O_3$ in the PRD-RAQMN Network. Obviously, concentrations of pollutants are smooth. The effect of very local emissions was not seen.

[Figure]

Fig.R2 Spatial distribution of average concentrations of $NO_2$ and $O_3$ in the PRD-RAQMN Network, figure is annual report of Pearl River Delta Regional Air Quality Monitoring Network in 2013 (https://www.epd.gov.hk/epd/sites/default/files//epd/english/resources_pub/publications/files/PRD_2013_report_en.pdf)

- Sites in EANET are mostly located in islands (Hedo, Ogasawara and Oki) and remote regions (Rishiri, Ochiishi, Yusuhara, Sado-seki, Happo). More information can be found in Ban et al. (2016).

As for $NO_2$ measurements, we agree that molybden convertors devices may cause errors. Ge et al. (2013) compared the measurements at an urban site in Beijing in summer by commercially standard chemiluminescence-based (called CL hereafter) instruments and Aerodyne Cavity Attenuated Phase Shift Spectroscopy (CAPS). The CAPS NO2 monitor directly measures the absorption of $NO_2$ at the wavelength of 450 nm and requires no conversion of $NO_2$ to other species.

Fig. R3-R4 presents the comparison between instruments. Generally, the biggest discrepancy appeared in 12:00-16:00, with a magnitude of 10-20%. In other periods, NO2 by CL and CAPS were similar. On average, discrepancies between CL and CAPS were less than 10%. The linear fitting slope reached 0.999 between CL and CAPS.

As shown in Fig. R4, observations between CL and CAPS agreed well with each other with hourly $NO_2$>15 ppbv. In low hourly $NO_2$(<10 ppbv), CL $NO_2$ overestimated CAPS by 10-30%. This is consistent with the statement by the reviewers, which

reported NO2 exist sometimes biases for stations far from sources in the measurements.

In this study, we compared observed monthly mean $NO_2$ with models, instead of daytime $NO_2$. This partly decreased the impact of errors from CL instrument. What's more, the observed NO2 in EA1 and EA3 were 20 ppbv or more. In these high NOx emission regions, biases from CL instruments may not bring too much impact on model validation. In EA4, most stations are located in islands or remote regions, with ~ 2 ppbv $NO_2$. The CL $NO_2$ will overestimated $NO_2$ concentrations.

In the revised manuscript, we added a discussion on observation sites and instruments in section 2.3.

[Figure]

Fig. R3 Observed mean diurnal variation of $NO_2$ in summer in Beijing by chemiluminescence-based (CL) instruments and CAPS in Beijing. Also shown is the difference of two instruments.

[Figure]

Fig. R4 Comparison of $NO_2$ measured by the CL NOx analyzer and CAPS.

**Comment 4**: I have the impression that authors do not need to include the EA2 region in the paper, you never use it in your discussions.

Reply: We agree. In the revised manuscript, we corrected it (EA1->EA1; EA3->EA2; EA4->EA3).

In this reply, we used EA1, EA3 and EA4 to give a clear comparison with the

previous manuscript.

**Comment 5**: Authors do evaluate several parameters relevant for model evaluation. It would have be better to have observations to put against models. It is often complicated to get all needed observations but maybe you can at list mention that in the prospectives. It become possible to have network ceilometers for PBLH evaluation. A lot of satellite observations are available to evaluate NOx or ozone at larger scales. What about vertical profiles?

Reply: We totally agree. In the revised manuscript, we collected observation data as much as possible. The new observation data includes:1) vertical profiles of $O_3$ in EA3 and EA4; 2) PBLH in EA1 and EA3; 3) dry deposition velocities in EA1 and EA4. We also discussed the model performance against these observations.

Fig. R5 presents the simulated and observed $O_3$ profiles in subregions. Because there was lack of $O_3$ sounding in EA1 in 2010, only observations in EA3 and EA4 are show. In general, ensemble means (Ense) presented an underestimation and overestimation for EA3 $O_3$ in middle (500-800 hpa) and lower (below 900 hpa) troposphere, respectively. In winter, the underestimation even extended to 200hpa in winter. The magnitudes of underestimation and overestimation reached 10-40 ppbv and 10-20 ppbv. In EA4, Ense reproduced the vertical structure of ozone in both summer and winter. An overestimation existed below 800 hpa in summer, with a magnitude of 10-20 ppbv.

[Figure]

Fig. R5 Simulated and observed $O_3$ profiles in summer and winter of 2010, averaged over all observed stations in three subregions over East Asia (EA1: left column, EA3: middle column, EA4: bottom column).

On dry depositions, most models underestimated dry deposition velocities of $O_3$ ($v_d$) in August-September, but still fell into the range of observed standard deviation. This partly explained the overestimation of $O_3$ concentrations in summer discussed in section 3.2. In October-November, simulated $v_d$ apparently overestimated observations by 30-50%.

In EA4, most stations were remote oceanic sites, and few dry deposition observations were conducted. So, we collected observations in other oceanic sites to evaluate model performance (Helmig et al., 2012). Tex, STR, GGSEX and AMMA represents observed ozone $v_d$ in (1) TexAQS06 (7 July–12 September 2006; north-western Gulf of Mexico), (2) STRATUS06 (9–27 October 2006; the persistent stratus cloud region off Chile in the eastern Pacific Ocean), (3) GasEx08 (29 February– 11 April 2008; the Southern Ocean), and (4) AMMA08 (27 April–18 May 2008; the southern and northern Atlantic Ocean). Because M11 $v_d$ were much higher than other models, we exclude M11 in calculating the Ense for $v_d$. As shown in Fig. R6, Ense of $v_d$ agreed with observations, reasonably. Both and simulated $v_d$ showed a July-September maximum.

[Figure]

Fig. R6 simulated and observed monthly $O_3$ dry deposition velocities. Observations in

EA1 were from Sorimachi et al. (2003) and Pan et al. (2010). Observations in EA4 were from Luhar et al. (2017).

Fig. R7 shows the comparison of simulated daytime PBL height with observations. In EA1, all the selected models exhibited the spring-maximum and winter-minimum season cycle, which captured the major pattern of climatology of PBLH observations (Guo et al.,2016). The Ense on PBLH was 100-200 m higher than radiosonde measurements. This is likely caused by the inconsistency of samples between models and measurements. The simulation was the mean value of 12 hours (08:00-20:00), while the average of measurements was calculated based on 3 hours (08:00, 14:00 and 20:00).

In EA3, observed PBLH did not varied as that in EA1, and differences between seasons were within 100 m. This pattern was captured by models. Similar as EA1, the simulated PBLH in EA3 was 100-200m higher than measurements.

Few measurements on remote oceanic site were conducted in East Asia. So, we compared simulations with European Centre for Medium-Range Weather Forecasts Reanalysis Data (von Engeln et al., 2013). Both showed a winter-maximum pattern of PBLH.

[Figure]

Fig. R7 Simulated daytime (08:00-20:00 LST) PBL height (m). Also shown are observed mean PBL height (m) at 08:00, 14:00 and 20:00 LST from Guo et al. (2016).

We totally agree with the reviewer that satellite observations evaluate NOx or ozone at larger scales. Sometimes satellite data is lack in cloudy or heavy haze days. So, the monthly values of satellite could not be averages of all days. Unfortunately, only monthly data of models (all days in one month) was submitted in MICS-Asia III. This inconsistency of samples between models and satellite would bring bias for model validation. So, we will conduct the model validation using satellite data in MCIS-Asia IV by collecting daily data.

Other comments etc...

**Comment 6**:Page 3-Line 7 – Please remind the value of the threshold

Reply: We added it (100 $\mu g/m^3$).

**Comment 7**:Page 10 - Line 4 – Please suppress "4)"

Reply: We deleted it.

**Comment 8**:Page 10 – Line 18 – A good example where using the ensemble average allows to better structure the discussion and to be more precise on the model skills.

Reply: We added a discussion on the using the ensemble average.

"In general, model results for three sub-regions exhibited a larger spread with a magnitude of 10-50 ppbv throughout the diurnal cycle than that in Europe and North America (Solazzo et al., 2012). The Ense $O_3$ in summer exhibited a systematic overestimation (20 ppbv) throughout the diurnal cycle in EA1. This indicated that models had difficulty dealing with $O_3$ in North China Plain. Compared with summer, there was only a slight systematic overestimation of Ense $O_3$ in other seasons (3-5 ppbv)"

**Comment 9**:Page 10– Line 24-25 – "...due to difficulties in dealing with vertical mixing": how do we know that?

Reply: In M11, the minimum of vertical diffusivity was set to be 0.5 $m^2$ $s^{-1}$. This value is a little higher than other models (e.g. CAMx: 0.1 $m^2$ $s^{-1}$).   In the stable boundary layer on nighttime, the higher vertical diffusivity may transport high ozone in upper layer to the surface, and also uplifted surface NO. The lower NO weakens the ozone titration.

    We realized that vertical mixing is not the only reason of nighttime ozone overestimation in M11. We needed more observed evidence to support our guess. So we deleted it in the revised manuscript.

**Comment 10**:Page 12 – Line 16 – How statistics are calculated? on hourly values?

Reply: These statistics are calculated by Appendix A in the revised manuscript based on monthly values. We added descriptions in the revised manuscript.

**Changes in the revised manuscript**: Page 21 **Appendix A. Statistical Measures**

**Comment 11**:Page 13– Line 16 – Why choosing a sub selection of models? It would be interesting to have all models.

Reply: We agree. It's better to present the intercomparison of PBLH from all models. Unfortunately, the other models have not outputted PBLH in this study. In MICS-Asia IV, all models will be requested to output PBLH.

**Comment 12**:Page 14 – Line 3 – Von Engeln no ?

Reply: Yes, it is "von Engeln".

**Comment 13**:Page 14 – Line 7 – You do not discuss VOC emissions. Would you suggest that models have no sensitivity to these emissions?

Reply: We plotted VOCs (ethene) emissions (Fig. R8). Compared with NO, the

consistency on ethene is better. Only M2 showed a small underestimation and overestimation in EA1 and EA3, respectively.

[Figure]

Fig.R8 NO (left) and ethene (right) emission fluxes on the first day in each month.

**Comment 14**: Page 14 – Line 15-20 – The discussion and the links between arguments are not that clear.

Reply: Thanks a lot.

"The difference in emissions allocations could contribute to the simulation variability. In the future, the projected gridded anthropogenic emissions should be provided to each group to eliminate the possibility that each group uses different mapping method. Interestingly, emissions in M1 and M8 exhibited similar levels, but their simulated $NO_2$, NO and $O_3$ presented a high intermodel variability in EA1 (Fig. 3 and Fig. 6). M1 simulated summer $O_3$ reached 80 ppbv while M8 was only 30 ppbv. This indicated that there were others causes to bring the intermodel variability on $O_3$."

**Comment 15**: Page 14 –Line 22 – I would say "net sink" since chemistry is a much higher absolute sink than deposition.

Reply: We agree.

**Comment 16**: Page 16 – Line 4 to 6 – Seems contradictory to have a small sink with considerable effect on oceanic surface. I would rather say that even if dry deposition velocities are small over oceanic surfaces, the impact of dry deposition over ocean is globally important because of the large surface ocean are representing.

Reply: We agree. In the revised manuscript, we reworded this sentence. "Compared to other regions, surface $O_3$ in EA4 were more sensitive to dry deposition parameterization

schemes in CTMs (Park et al.,2014). Park et al. (2014) revealed that O₃ on oceans differed by 5-15 ppbv in East Asia resulting from different dry deposition parameterization schemes". We deleted "Ganzeveld et al. (2009) revealed that surface O₃ may differ by up to 60% when O₃ dry deposition velocity varied from 0.01 to 0.05 cm/s."

**Comment 17:** Page 16 – Line 6-8 – Why can we do the assumption that dry deposition is specifically important for EA4?

Reply: This assumption was taken from Park et al. (2014), in which the impact of O₃ dry deposition was examine over East Asia. They found that O₃ mixing ratios in EA4 were more sensitive to dry deposition parameterization schemes in CTMs than other regions. O₃ decrease as low as 5-15 ppbv at stations in EA4 in Wesely scheme than M3DRY scheme (1990). In EA1 and EA3, the changes of O₃ only ranged from 0-5 ppbv.

**Comment 18:** Page 17 – Line 1 – I observe that range of concentrations for O3 and NOx can be very different between models but it is not clear if slopes are that different.

Reply: We plotted the slopes between NOₓ and O₃ in Fig. 8 in the revised manuscript. The slopes between NOₓ and O₃ in EA1 ranged from -2.84 to -0.09 between models.

[Figure]

Fig. R9 Scatter plots between monthly daytime (08:00-20:00) surface NOₓ and O₃ at each station over EA1(red), EA3(green)and EA4(blue) in May-October, for observations(obs) and models

**Comment 19:** Page 18 – Line 2 to 5 but also Line 7 to 20 – The variability authors are

mentioning is not clear from figure 9. Also for differences between winter and summer, we need to have numbers to better evaluate this variability.

Reply: Thanks. Line 2-5: "A small variability in winter appeared below 900 hPa in three sub-regions, and slowly decreased with height. The mean standard deviation ($\sigma$) below 900 hpa were 7.6 ppbv, 6.9 ppbv and 6.0 ppbv in EA1, EA3 and EA4, which covered 18.3%, 15.0% and 15.4% of mean $O_3$ concentrations. In 700-900 hpa, $\sigma$ decreased to 5.4 ppbv, 4.4 ppbv and 4.8 ppbv in EA1, EA3 and EA4, 12.2%, 9.4% and 10.8% of mean $O_3$ concentrations".

Line 7-20: "With the increase of solar radiation and air temperature, vertical profiles were more scattered in the lower troposphere in summer. In polluted regions (EA1), various vertical structures of $O_x$ were found below 700 hPa. $\sigma$ reached 16.3 ppbv, 20.8 % of mean concentrations, which was higher than winter (6.2 ppbv, 15.2%). … In EA3, vertical structures of Ox among models were consistent, but concentrations differed more than those in EA1. The mean standard deviation of models covered 22% of mean concentrations".

Table R3 Ensemble mean simulated ozone (Ense) and its standard deviation(std) in EA1

|  | Winter | | | Summer | | |
|---|---|---|---|---|---|---|
|  | Ense/ppbv | Std/ppbv | Std/Ense (%) | Ense/ppbv | Std/ppbv | Std/Ense(%) |
| 1000-900 hpa | 41.4 | 7.6 | 18.3 | 82.1 | 17.7 | 21.6 |
| 900-700 hpa | 44.3 | 5.4 | 12.2 | 78.4 | 14.2 | 18.1 |
| 700-550 hpa | 51.3 | 7.0 | 13.5 | 70.1 | 11.7 | 16.7 |
| 550-300 hpa | 87.0 | 82.8 | 95.2 | 89.4 | 30.6 | 34.2 |

**Comment 20**: Page 18 – Line 5-6 – Authors do have this information, it should more than an suggestion, no?

Reply: Thanks a lot. This sentence is our guessed possible causes and we have not more evidences on the impact of convection and turbulent mixing on vertical profiles. So we deleted this sentence in the revised manuscript. In the MICS-Asia IV, we will directly output the impact of each process (convection, turbulent) from all models.

**Comment 21**: Page 19 – Line 8 – 9 – Itis mention that dispersion between models is higher here than for the European case and authors suggest the models do not represent uncertainties, could you develop? Also authors mention that key processes could miss, what kind of processes are they thinking to?

Reply: Thanks a lot. We totally agree that an ensemble averages representing the uncertainty of $O_3$ is helpful. In MICS-ASIA III, the arithmetic means of all models is difficult meet this criteria, although it has been successfully in other regions. Potempski and Galmarini (2009) did some basic theoretical to find optimal linear combination of

model results with the help of complex mathematical tools. Solazzo et al. (2012) used this method for O₃ ensemble in Europe and North America. They found that the most skillful ensemble is not necessarily generated by including all available models, and suggested that the clustering technique could generate a better ensemble average, but needs further refinement. This is beyond the scope of this manuscript and will be the major topic of our next manuscript

We mentioned that most models did not reflect this uncertainty or missed key processes in MICS-Asia III. The parameterization of heterogeneous chemistry in models is possibly a key process. The manuscript by Akimoto et al. (2019) in this special issue found that the missing heterogeneous "renoxification" reaction of $HNO_3$ on soot in most models except NAQPMS would partly explain the overestimation of simulated $O_3$ mixing ratios. The treatment of $O_3$ vertical transport in models also affect the simulated results significantly in Akimoto et al. (2019).

**Comment 22**:Page 20 – Line 11 to 15 – Do we observe same differences for higher levels? Maybe in some models plumes are also present but at different altitudes.

Reply: We also compared simulated O3 in upper boundary layer (Fig. R10). The results were similar as surface ozone.

[Figure]

Fig. R10 500m O₃ spatial distribution from 13 models for summer 2010

**Comment 23**:Page21 – Line 2 – I'm not sure that author do define mathematically the coefficient of variation.

Reply: The CV is defined as the standard deviation of the modeled fields divided by the average. The larger the value of CV, The lower the consistency among the models.

**Comment 24**:Page 21 – Line 13 – Like in table1 authors do mention that "default" is used as boundary conditions. Default values should be more clearly defined? climatology? from where?

Reply: In MICS-ASIA III, M2 and M7 made boundary conditions depending on their own previous experience denoted by "default" in Table 1.

In M2, the default initial condition and boundary conditions were based on Gipson (1999) to represent the clean air concentrations, and have been formulated from available measurements and results obtained from modeling studies.

In M7, the default initial condition and boundary conditions were derived from the idealized profile based upon northern hemispheric, mid-latitude, clean environment conditions from a NOAA-Aeronomy Laboratory Regional Oxidation Model (NALROM) (Liu et al.,1996).

**Comment 25**:Page 22 –Line 7 – " .. its relevant species .." I also see VOC or even radicals as relevant species for the tropospheric ozone cycle then it is better to mention 03 and NOx instead.

Reply: We agree and revised it.

**Comment 26**:About Table and Figures Table2 – Maybe it is mandatory to mention how statistic alindicator are calculated (i.e formula). Be careful "suqare" in the title instead of square. RMSE do have units, please mention it. Figure 1 – as mention earlier I would have removed EA2 that is not discussed.

Reply: We agree. We listed the formula in the Appendix A in the revised manuscript. And also added RMSE units and corrected "suqare" to "square". In the revised manuscript, we removed EA2.

**Comment 27**:Figure 2 – probably too small as it is. The full blackline does not seems necessary.

Reply: We revised it.

**Comment 28**:Figure 9 – Maybe it is possible to reduce horizontal scale down to 10 ppb to have more space on the right and to better evaluate the ensemble dispersion.

Reply: We revised it.

**Comment 29**:Figure 10 – Maybe too small also Figure 11 – Same as Figure10

Reply: We revised it.

References:

Akimoto, H., Nagashima, T., Li, J., Fu, J. S., Ji, D., Tan, J., and Wang, Z.: Comparison of surface ozone simulation among selected regional models in MICS-Asia III – effects of chemistry and vertical transport for the causes of difference, *Atmos. Chem. Phys*., 19, 603-615, https://doi.org/10.5194/acp-19-603-2019, 2019.

Ban, S. , Matsuda, K. , Sato, K. , Ohizumi, T. : Long-term assessment of nitrogen deposition at remote EANET sites in japan. *Atmos. Environ*, 146, 70-78, 2016.

Ge, B., Sun, Y., Liu, Y., Dong, H., Ji, D., Jiang, Q., Li, J., and Wang, Z.: Nitrogen dioxide measurement by cavity attenuatedphase shift spectroscopy (CAPS) and implications in ozone production efficiency and nitrate formation in Beijing, China, *J. Geophys. Res. Atmos*., 118, doi:10.1002/jgrd.50757,2013.

Gipson, G. L.: The Initial Concentration and Boundary Condition Processors. In Science algorithms of the EPA Models-3 Community Multiscale Air Quality (CMAQ) Modeling System, US Environmental Protection Agency Report, EPA-600/R-99/030, 12-1–12-91, 1999.

Guo, J., Miao, Y., Zhang, Y., Liu, H., Li, Z., Zhang, W., He, J., Lou, M., Yan, Y., Bian, L., and Zhai, P.: The climatology of planetary boundary layer height in China derived from radiosonde and reanalysis data, *Atmos. Chem. Phys*., 16, 13309-13319, https://doi.org/10.5194/acp-16-13309-2016, 2016

Helmig, D., Lang, E. K., Bariteau, L., Boylan, P., Fairall, C. W., Ganzeveld, L., Hare, J. E., Hueber, J., and Pallandt, M.: Atmosphere-ocean ozone fluxes during the TexAQS 2006, STRATUS 2006, GOMECC 2007, GasEx 2008, and AMMA 2008 cruises, *J. Geophys. Res*., 117, D04305, doi:10.1029/2011JD015955, 2012.

Li, J., Yang, W., Wang, Z., Chen, H., Hu, B., Li, J. J., Sun, Y., Fu, P., Zhang, Y..: Modeling study of surface ozone source-receptor relationships in East Asia. *Atmos. Res.,* S0169809515002227, 2015.

Liu, S. C., McKeen, S. A., Hsie, E-Y., Lin, X., Kelly, K. K., Bradshaw, J. D., Sandholm, S. T., Browell, E. V., Gregory, G. L., Sachse, G. W., Bandy, A. R., Thornton, D. C., Blake, D. R., Rowland, F. S., Newell, R., Heikes, B. G., Singh, H., and Talbot, R. W. : Model study of tropospheric trace species distributions during PEM-West A*, J. Geophys. Res.,* 101, 2073-2085,1996.

Zhong, L., Louie, P., Zheng, J., Wai, K. M., Josephine W.K. Ho, Yuan, Z., Lau, A. K. H., Yue, D., Zhou, Y.: The Pearl River Delta Regional Air Quality Monitoring Network – Regional Collaborative Efforts on Joint Air Quality Management, Aerosol

and Air Quality Research, 13: 1582–1597, 2013

Pan, X., Wang Z., Wang X., Dong H., Xie, F., Guo, Y.: An observation study of ozone dry deposition over grassland in the suburban area of Beijing. Chinese Journal of Atmospheric Sciences (in Chinese), 34(1), 120-130, 2010.

Park, R. J., et al. : An evaluation of ozone dry deposition simulations in East Asia. Atmospheric Chemistry and Physics 14(15): 7929-7940,2014.

Potempski, S., Galmarini, S.: Est Modus in Rebus: analytical properties of multi-model ensembles. Atmospheric Chemistry and Physics 9, 9471-9489,2009.

Solazzo, E., Bianconi, R., Pirovano, G., Matthias, V., Vautard, R., Appel, K. W., Bessagnet, B., Brandt, J., Christensen, J. H., Chemel, C., Coll, I., Ferreira, J., Forkel, R., Francis, X. V., Grell, G., Grossi, P., Hansen, A., Miranda, A. I., Moran, M. D., Nopmongco, U., Parnk, M., Sartelet, K. N., Schaap, M., D. Silver, J., Sokhi, R. S., Vira, J., Werhahn, J., Wolke, R., Yarwood, G., Zhang, J., Rao, S. T., Galmarin, S.: Model evaluation and ensemble modelling of surface-level ozone in Europe and north America in the context of AQMEII, *Atmos. Environ*, 53(6), 60-74,2012.

Sorimachi, A, Sakamoto, K, Ishihara H, Fukuyama, T., Utiyama, M., Liu, H., Wang, W., Tang, D., Dong, X., Quan, H.: Measurements of sulfur dioxide and ozone dry deposition over short vegetation in northern China-A preliminary study. *Atmos. Environ*., 37(22), 3157-3166, 2003.

Zheng, J., Zhong, L. , Wang, T. , Louie, P. K. K. , Li, Z.: Ground-level ozone in the pearl river delta region: analysis of data from a recently established regional air quality monitoring network. *Atmos. Environ*, 44(6), 814-823,2010.

---

## Author Response (AR2)

We thank the reviewer for his/her constructive comments.
Response to the Specific comments.

**Comment 1:** What does the vertical axis on the right of Fig. R1 mean?
If it represents a 3 km mesh emission, contrary to what is stated in the response document or in the revised manuscript, it is likely that the observation points receive local emissions.

**Reply:** Before the intercomparison between emission rates in 45 km and 3 km emission inventory, we provide the equation how to calculate the emission rates ($\mu g/m^2/s$) in 45 km resolution ($E_{45km}$) from those in 3 km resolution ($E_{3km}$).

$$E_{45km} = \sum_{i=1}^{m=15} \sum_{j=1}^{n=15} (E_{3km\ ij} /(m \times n)) \quad (1)$$

As shown in Eq (1), some $E_{3km\ ij}$ were higher than $E_{45km}$. Others could be less than $E_{45km}$. In general, there could be local emissions in $i^{th}$ and $j^{th}$ grid when $E_{3km}$ in this grid is much higher than $E_{45km}$. When $E_{3km}$ is much less than $E_{45km}$, the $i^{th}$ and $j^{th}$ grid represented a regional background condition. If $E_{3km}$ is close to $E_{45km}$, the emission rates in the $i^{th}$ and $j^{th}$ grid represented the averaged conditions of 45 km $\times$ 45 km areas. So, the comparison of $E_{3km}$ with $E_{45km}$ is a good way to examine if this 3km grid receives local emissions.

[Figure]

Fig. R1 The grids of observations in 45 (left) and 3 km (right) emissions inventory. The solid cycle represents the location of observation site.

In this study, we calculated the NO emission rates of observation sites in 45 km grid ($E_{45km}$). Emission rates ($E_{3km}$) at each 3 km grid (i=1,15; j=1,15) within this 45km grid was compared with $E_{45km}$. Meanwhile, Emission rates ($E_{3km\ obs}$) at this 3 km grid of

observation site was shown. Fig. R2 and R3 showed the comparison of $E_{45km}$, $E_{3km}$ and $E_{3km\ obs}$ for NO and $C_2H_4$ emission rates. Clearly, $E_{3km\ obs}$ at the observation sites were close to $E_{45km}$, which indicated that these observation sites rarely receive local emissions.

[Figure]

Fig. R2 Scatter plots of $E_{45km}$, $E_{3km}$ and $E_{3km\ obs}$ for NO emission rates ($\mu g/m^2/s$). The gray and red solid cycles represented the relationships of $E_{45km}$-$E_{3km}$ and $E_{45km}$- $E_{3km\ obs}$. The gray solid cycles above the line y=2x represented the grids receive the local emissions.

Fig. R3 The same as Fig.2, but for ethene

The following is a comment on the revised manuscript other than the above.

**Comment 2:** Page 11 Line 11: It describes the measuring instruments, but which measuring stations adopt these measuring instruments?
**Reply:** In this study, $NO_x$ was measured by Thermo Scientific 42C NO-$NO_2$-$NO_x$

Analyzer with chemiluminescence technology at 40 sites in all three network (CERN, PRD-RAQMN). $O_3$ were measured by Thermo Scientific 49i with UV photometric technology from USA in CERN network and by Thermo Scientific 49C in PRD-RAQMN and EANET network.

**Changes in the revised manuscript**: Page 8 Line 14-17.

**Comment 3**:<About the response of my Comment 3>
It is said that the ensemble of models underestimates the measured values of dry deposition velocity from August to September in EA1, but according to Figure R3 it is actually overestimated.
Therefore, the claims in the response document or in the revised manuscript that underestimation of dry deposition velocity contributed to overestimation of summer ozone concentrations are false and need to be reviewed.
**Reply:** We agree. In the revised manuscript, we reworded related discussions and conclusions.
 "In EA1, ensemble mean values overestimated observed dry deposition velocities of $O_3$ (vd) in August-September, but still fell into the range of observed standard deviation. This indicated that there must be other factors rather than dry deposition playing important roles in the overestimation of August-September $O_3$ in EA1. In October-November, simulated vd apparently underestimated observations by 30-50%. Among models, the lower dry deposition velocities in May-July from M1, M2, M4 and M6 than that of M11 partly explained higher May-July surface O3 from those simulations than that from M11. However, M13 and M14 still produced high O3 concentrations in May-September although their dry deposition velocities were similar to that of M11(Fig. 3)."

**Changes in the revised manuscript**: Page 17 Line 14-23.

The following are comments on the revised manuscript other than the above.

**Comment 4**:Section 3.5: The sentences are difficult to understand. I think that English native check is necessary first.
**Reply:** We invited an English native speaker to check our manuscript through AJE company.

**Comment 5**:Page 16 Line 4-5: I do not understand the meaning of this sentence.
**Reply:** We revised to ". Quantifying the contributions of these processes can help explain model biases through sensitivity simulations"

**Changes in the revised manuscript**: Page 16 Line 24-25.

**Comment 6**:Page 16 Line 27 "summer": A statement from May to July, not summer,

is more logical.

**Reply:** We agree and revised it.

**Comment 7**:Page 17 Line 3-4: According to Fig.9, contrary to the description in the revised manuscript, the model underestimated the observed values of Vd.

**Reply:** We agree and corrected it.

**Changes in the revised manuscript**:   Page 16 Line 24-25.

**Comment 8**:Page 18 Line 7-8: The description in this sentence should indicate that it is a description of the observed value.

**Reply:** We agree.   "The slope and intercept of regression line between observed $O_3$ and NOx were -0.77 ppbv/ppbv and 59.5 ppbv, respectively"

**Changes in the revised manuscript**:   Page 18 Line 27.

**Comment 9**:Page 18 Line 16: What are "previous theoretical results"?

**Reply:** According to comments by co-editor, we deleted these sentences.

**Comment 10**:Page 19 Line 15-25: At the beginning of chapter 5, you have mentioned that you evaluated dry deposition, PBL, and chemistry.

I do not understand why the discussion about the meteorological field comes out suddenly here.

**Reply:** In the revised manuscript, we remove this discussion.

**Comment 11**:<About the response of my Comment 6>

Page 5 Line 9-10: This sentence is about how to validate the model, so this sentence should be moved to chapter 3 which discusses it.

**Reply:** We moved it to section 3.2.

**Changes in the revised manuscript**:   Page 9 Line 25.

**Comment 12**:<About the response of my Comment 9>

Page 7 Line 2-6: Make a description that shows the correspondence between the model and the university.

**Reply:** We agree. "GEOS-Chem was run with a 2.5º×2º horizontal resolution and 47 vertical layers by University of Tennessee and Chemical AGCM for Study of Atmospheric Environment and Radiative Forcing (CHASER) was run with a 2.8º× 2.8º

horizontal resolution and 32 vertical layers by and Nagoya University."

**Changes in the revised manuscript**: Page 7 Line 6-8.

**Comment 13**: <About the response of my Comment 17>

It is understood from Appendix A that various statistics are derived taking into account only the variation due to the location (i) of the measuring station, but in fact it seems that temporal variations are also taken into consideration.
This is because, for example, in Page 11 Line 22-24, as the reason for the high correlation, the high reproducibility of the monthly variation is mentioned.
**Reply:** We totally agree. We added the variation due to temporal variations in Appendix A.

**Comment 14**: <About the response of my Comment 27>

I understood that all models must be identified in Fig.2 and Fig.3.
However, since it is still difficult to identify models by color, I hope that at least models that appear in the discussion in chapter 5 can be identified by such as drawn lines.
**Reply**: We agree. In Fig. 3 and 4 in the revised manuscript, we only plotted models in appear in the discussion in chapter 5 by colors. The others are plotted in gray lines.

<Other technical corrections to the revised manuscript>

As there are many necessary technical corrections, I would like you to carefully review the entire manuscript. .
Some of them are shown below.

**Comment 15**: Page 2 Line 2: "Evaluated and intercompared to O3 observations" perhaps should be "intercompared and evaluated to O3 observations".
**Reply:** We revised it.

**Comment 16**: page 2 Line 7-8: "western pacific rim" is repeated twice.
**Reply:** We deleted the second one.

**Comment 17**: Page 8 Line 25: "Table1" should be "Table2".
**Reply:** We revised it.

**Comment 18**: Page 11 Line 27: "Table1" should be "Table2".

**Reply:** We revised it.

**Comment 19**:page 14 line 27: The word "other" should be added before "combined influence".
**Reply:** We revised it.

**Comment 20**:Page 17 Line 24 "Sillman and He et al.": "Et al" should be removed.
**Reply:** We revised it.

**Comment 21**:Page 20 Line 10-11: There is an incomplete sentence.
Some of figures: "EA3" should be replaced by "EA2", and "EA4" should be replaced by "EA3".
**Reply:** We revised it. "For the North China Plain and western Pacific Rim, the model ensemble severely overestimated surface O3 levels for May-September by 10-30 ppbv."

**Changes in the revised manuscript**: Page 23-25

We thank co-editor for his constructive comments.
Response to the Specific comments.

**General comment:** Although the manuscript is improved, still many fundamental and technical points are present needing revision. Please examine comments from the reviewer #1 carefully, and the following comments by the Co-Editor. Recheck English throughout the manuscript with a native speaker.

**Reply:** Thanks a lot for insightful comments. In the revised manuscript, we invited a native speaker by AJE company to improve our language in this manuscript.

(pages and lines are for the change-track version)

**Comment 1:** Page 2, line 18: In terms of
**Reply:** We revised it.

**Comment 2:** Page 2, line 20: hPa
**Reply:** We revised it.

**Comment 3:** Page 3, line 21: do not
**Reply:** We revised it.

**Comment 4:** Page 5, lines 19-20: Better to place the following inserted sentence elsewhere (page 8-9?): If two or more observation sites were in the same grid of model, their mean values will be used to evaluate model performance.
**Reply:** We placed it into Page 9 Line 25.

**Comment 5:** Page 6, line 18. possibly
**Reply:** We revised it.

**Comment 6:** Page 8, lines 18-20: More explanation and clarification about the local sources and sinks are necessary, as requested by the reviewer. Not only for NOx but also VOC, representativeness is necessary.
**Reply:** We agree. In the revised manuscript, we plotted NO and VOCs emission rates.

Before the intercomparison between emission rates in 45 km and 3 km emission inventory, we provide the equation how to calculate the emission rates ($\mu g/m^2/s$) in 45 km resolution ($E_{45km}$) from those in 3 km resolution ($E_{3km}$).

$$E_{45km} = \sum_{i=1}^{m=15} \sum_{j=1}^{n=15} (E_{3km\,ij} / (m \times n)) \quad (1)$$

As shown in Eq (1), some $E_{3km\,ij}$ were higher than $E_{45km}$. Others could be less than $E_{45km}$. In general, there could be local emissions in $i^{th}$ and $j^{th}$ grid when $E_{3km}$ in this grid is much higher than $E_{45km}$. When $E_{3km}$ is much less than $E_{45km}$, the $i^{th}$ and $j^{th}$ grid represented a regional background condition. If $E_{3km}$ is close to $E_{45km}$, the emission

rates in the $i^{th}$ and $j^{th}$ grid represented the averaged conditions of 45 km×45 km areas. So, the comparison of $E_{3km}$ with $E_{45km}$ is a good way to examine if this 3km grid receives local emissions.

[Figure]

Fig. R1 The grids of observations in 45 (left) and 3 km (right) emissions inventory. The solid cycle represents the location of observation site.

In this study, we calculated the NO emission rates of observation sites in 45 km grid ($E_{45km}$). Emission rates ($E_{3km}$) at each 3 km grid (i=1,15; j=1,15) within this 45km grid was compared with $E_{45km}$. Meanwhile, Emission rates ($E_{3km\ obs}$) at this 3 km grid of observation site was shown. Fig. R2 and R3 showed the comparison of $E_{45km}$, $E_{3km}$ and $E_{3km\ obs}$ for NO and $C_2H_4$ emission rates. Clearly, $E_{3km\ obs}$ at the observation sites were close to $E_{45km}$, which indicated that these observation sites rarely receive local emissions.

[Figure]

Fig. R2 Scatter plots of $E_{45km}$, $E_{3km}$ and $E_{3km\ obs}$ for NO emission rates ($\mu g/m^2/s$). The gray and red solid cycles represented the relationships of $E_{45km}$-$E_{3km}$ and $E_{45km}$- $E_{3km\ obs}$. The gray solid cycles above the line y=2x represented the grids receive the local emissions.

[Figure]

Fig. R3 The same as Fig.2, but for Ethene

**Comment 8:** Page 8, line 28: exhibited
**Reply:** We revised it.

**Comment 9** Page 9, line 1: Molybdenum converters
**Reply:** We revised it.

**Comment 10**:Page 9, lines 2-5. The results must be dependent on the conditions (with different NOz/NOx). Where did the comparison take place? Uncertainty should be estimated to cover all possible conditions (urban to rural) used for the study.

**Reply:** We agree. In this study, the comparison was conducted in an urban site in Beijing from 1 to 29 August 2012. As said by the editor, the difference between $NO_2$ and $NO_2^*$ is dependent on the conditions (Fig. R1). In general, their differences appeared within 10-15% when $NO_2^*$>15 ppbv and NOz<10 ppbv in Beijing. In low $NO_2$ (<15 ppbv) and high NOz (>10 ppbv) conditions, the $NO_2^*$ usually was higher than $NO_2$ (10-30%). Unfortunately, there are very few CAPS $NO_2$ measurements in China. Jung reported their $NO_2$ and $NO_2^*$ measurements in a rural site in South Korea (Fig. R2). Similar as Beijing, $NO_2$ and $NO_2^*$ usually appeared within 20% when $NO_2^*$ >20 ppbv in Spring and Summer. $NO_2^*$ overestimated $NO_2$ by 20-40% when $NO_2^*$ <20 ppbv. In Fall and Winter, $NO_2^*$ usually overestimated $NO_2$ by 10-20 ppbv in all conditions.

In the revised manuscript, we added a short discussion on $NO_2^*$ uncertainty.

"This bias was dependent on the chemical conditions. A one-month continuous measurement in August by a chemiluminescence analyzer and Aerodyne Cavity Attenuated Phase Shift Spectroscopy (CAPS) at an urban site in Beijing showed that this bias from a chemiluminescence analyzer was small when NO2 concentrations were more than 10-15 ppbv, and ranged from 10% to 30% under low NO2 (<10 ppbv) (Ge et al., 2013). Measurements at a rural site in South Korea also revealed a similar pattern (Jung et al., 2017). These comparisons suggested that observations by molybdenum converters may overestimated NO2 by 10-20% in EA1 and EA2, and 30% in EA3. This brings uncertainties for the model evaluation on $NO_2$ in this study. "

[Figure]

Fig. R1 Comparison of hourly $NO_2$ measured by the CL $NO_x$ analyzer and the CAPS $NO_2$ monitor in Beijing

[Figure]

Fig. R2 The $NO_2/NO_2*$ ratios as a function of $NO_2*$ concentrations during (a) fall, (b) winter, (c) spring, and (d) summer in a rural site in South Korea (Jung et al., 2017). $NO_2$ and $NO_2*$ is the measurements by the photolytic converterver and molybdenum converter。

**Changes in the revised manuscript**:    Page 8 Line 20-27.

**Comment 11**:Page 9, line 16: parts per billion by volume
**Reply:** We revised it.

**Comment 12**:Page 9, line 19: and M7
**Reply:** We revised it.

**Comment 13**:Page 9, line 20: closer to
**Reply:** We revised it.

**Comment 14**:Page 9, line 20: M11 simulated $O\_3$ with RMSEs of
**Reply:** We revised it.

**Comment 15**:Page 12, line 26: existed
**Reply:** We revised it.

**Comment 16**:Page 13, line 21. approximately 20 and 10 ppbv
**Reply:** We revised it.

**Comment 17**:Page 14, line 7: showed a better performance

**Reply:** We revised it.

**Comment 18**:Page 15, line 5: What is meant by short-lived species?
**Reply:** This underestimation partly was attributed to the coarse model horizontal resolution (45km) used in the MICS-Asia III, which hardly reproduced concentrations of short-lived species (e.g. NO).

**Comment 19**:Page 15, line 25: Figure 5 shows
**Reply:** We revised it.

**Comment 20**:Page 15, lines 27-28 and elsewhere: hPa
**Reply:** We revised it.

**Comment 21**:Page 16, line 7: The mean standard deviations (SD) of models ($1\sigma$) Use SD instead of $\sigma$ later.
**Reply:** We revised it.

**Comment 22**:Page 23, line 16. MICS
**Reply:** We revised it.

**Comment 23**:Page 27, lines 23. Ensemble mean values rather OVERESTIMATES vd (Fig. 9). Discussion needs to be revised (conclusion also). Representativeness of deposition velocity observations over the domains needs to be discussed. I doubt that grassland represents the whole domain region.
**Reply:** We agree. We reworded discussions (and conclusions) on dry depositions in the revised manuscript.
"In EA1, ensemble mean values overestimated observed dry deposition velocities of O3 (vd) in August-September, but still fell into the range of observed standard deviation. This indicated that there must be other factors rather than dry deposition playing important roles in the overestimation of August-September $O_3$ in EA1. In October-November, simulated vd apparently underestimated observations by 30-50%. Among models, the lower dry deposition velocities in May-July from M1, M2, M4 and M6 than that of M11 partly explained higher May-July surface $O_3$ from those simulations than that from M11. However, M13 and M14 still produced high $O_3$ concentrations in May-September although their dry deposition velocities were similar to that of M11."

In this study, we selected observations at two sites in EA1. One site was located in a valley of Mangshan Forest Park, the other (CREAS) was in a suburb about 10km north of the Beijing city. The CREAS site had an area of 200m×200m, and was thickly covered with short grass about 10cm high (Dense grassland). Fig.R3 presents the land cover classification map in EA1 from MODIS satellite data (Zhang et al., 2008). It can be found that dense grassland is one of the dominated landcover classes and covers ~20% land area. Another dominated landuse in EA1 is crop class, which covers more

than 50% land area. Unfortunately, there are few observations on $O_3$ dry deposition on the crop class in China. This may bring uncertainties for model evaluation. Hardacre et al. (2015) reported $O_3$ dry deposition measurements on crops in Europe and simulated $O_3$ dry deposition in 15 global models. Both observations and simulations showed that $O_3$ dry deposition velocities on agriculture crop class were quite similar as grasslands (Fig. R4). This indicated that the uncertainties on representativeness of measurement sites in this study did not affect our conclusions.

We added related discussions in the revised manuscript.

**Changes in the revised manuscript**:    Page17 Line 25-Page 18 Line 2.

[Figure]

Fig. R3 The land cover classification map from MODIS satellite data

[Figure]

Fig. R4 Total annual $O_3$ dry deposition and annual average $O_3$ deposition velocity partitioned to land cover classes (GL: grassland; AC: agriculture crop class)

**Comment 24**: . Page 28, lines 3-4. October-November, simulated vd apparently overestimated observations by 30-50%. This is also opposite.

**Reply:** We agree. In the revised manuscript, we corrected it. "In October-November, simulated vd apparently underestimated observations by 30-50%."

**Changes in the revised manuscript**:    Page 17 Line 18.

**Comment 25**:Page 29, line 6. Is this only because O3 is titrated with NO? The analysis never tells the regimes. Consider cases of O3 buildup and transport by consuming NOx. Even in the NOx limited regime, the NOx-O3 relationship constructed from geographical distribution will show a negative correlation. At least potential ozone ($O_3$ + $NO_2$) should be used for the analysis, and remove all statements on the regimes (particularly those in lines 22-23, page 29).

**Reply:** We agree. In the revised manuscript, we remove all statements on the regimes.

Fig. R5 shows the relationship between $NO_x$ and $O_x$($O_3+NO_2$) in this study. For EA1, observed $O_x$ increases with the increase of $NO_x$ levels, with coefficient of determination ($R^2$) of 0.61. Most of the models (except for M8, M11 and M13) failed to reproduced observed positive correlations between $O_x$ and $NO_x$, and their $R^2$ only ranged from 0.01-0.08. The slope, intercept and $R^2$ of M8 and M11 are relative agreement with observations. For EA2, all models reproduced observed key patterns in which $O_x$ positively correlated with $NO_x$.

[Figure]

Fig. R5 Scatter plots for monthly daytime (08:00-20:00) surface $NO_x$ and $O_x$ for each station in EA1 (red), EA2 (green)and EA3 (blue) in May-October, for observations (obs) and models. Also shown are the linear regression equations and coefficient of determination ($R^2$) for $NO_x$ and $O_x$ ($O_3+NO_2$) in EA1 (red) and EA2 (green).

**Changes in the revised manuscript**:    Page 19 Line 7-15.

**Comment 26:** Page 30, line 6. Briefly mention how PBLH was measured.

**Reply:** In this study, the observed PBLH was derived from the radiosonde network of the L-band sounding system of the China Meteorological Administration (CMA). The system provides fine-resolution profiles of temperature, pressure relative humidity, wind speed and direction. the bulk Richardson number (Ri) method (Vogelezang and Holtslag,1996) was taken to simultaneously estimate the PBLH from CMA soundings. $R_i$ is defined as the ratio of turbulence associated with buoyancy to that induced by mechanical shear, which is expressed as

$$Ri(z) = \frac{(g/\theta_{vs})(\theta_{vz} - \theta_{vs})(z - z_s)}{(u_z - u_s)^2 + (v_z - u_s)^2 + (bu_*^2)},$$

where z denotes height above ground, s the surface, g the acceleration due to gravity, θv virtual potential temperature, u and v the component of wind speed and u∗ the surface friction velocity. u∗ can be ignored here due to the much smaller magnitude compared with bulk wind shear term in the denominator (Vogelezang and Holtslag, 1996). The critical value of 0.25 (Ri) is referred to as PBLH in this study, similar to the criteria used by Seidel et al. (2012).

**Changes in the revised manuscript**: Page 19 Line 27-Page 20 Line 5.

**Comment 27:** Page 30, line 8. caused by the sampling bias between
**Reply:** We revised it.

**Comment 28:** Page 31, line 9. A model ensemble was produced
**Reply:** We revised it.

**Comment 29:** Page 31, line 14. A period character must be a comma?
**Reply:** We revised it.
"In North China Plain and western Pacific rim, the model ensemble severely overestimated surface $O_3$ in May-September by 10-30 ppbv."

**Comment 30:** At many times figures contain EA1, 3, and 4, instead of EA1, 2, and 3.
**Reply:** We corrected it.

Jinsang Jung, JaeYong Lee, ByungMoon Kim, SangHyub Oh, Seasonal variations in the NO2 artifact from chemiluminescence measurements with a molybdenum converter at a suburban site in Korea (downwind of the Asian continental outflow) during 2015–2016, Atmospheric Environment, Volume 165,2017,Pages 290-300.

Seidel, D. J., Zhang, Y., Beljaars, A., Golaz, J.-C., Jacobson, A. R., and Medeiros, B.: Climatology of the planetary boundary layer over the continental United States and Europe, J.Geophys.Res.Atmos., 117, D17106, doi:10.1029/2012JD018143, 2012

Vogelezang, D. H. P. and Holtslag, A. A. M.: Evaluation and model impacts of alternative boundary-layer height for mulations, Bound.-Lay. Meteorol., 81, 245–

269, doi:10.1007/BF02430331, 1996.

Xia Zhang, Rui Sun, Bing Zhang, Qingxi Tong, Land cover classification of the North China Plain using MODIS_EVI time series, ISPRS Journal of Photogrammetry and Remote Sensing, Volume 63, Issue 4,2008,Pages 476-484.

[revised manuscript text omitted]

*Fig.1 Li et al., 2018*

[Figure]

*Fig.2 Li et al., 2018*

[Figure]

[Figure]

*Fig.3 Li et al., 2018*

[Figure]

[Figure]

*Fig.4 Li et al., 2018*

[Figure]

[Figure]

*Fig.5 Li et al., 2018*

[Figure]

*Fig.6 Li et al., 2018*

.

[Figure]

*Fig.7 Li et al., 2018*

[Figure]

*Fig.8 Li et al., 2018*

[Figure]

*Fig.9 Li et al., 2018*

[Figure]

[Figure]

*Fig.10 Li et al., 2018*

---

## Author Response (AR3)

We thank the reviewer for his/her constructive comments.
Response to the Specific comments.

About the reply to my Commnet1>
**Comment 1:** Thank you for the detailed explanation on the investigation of the spatial representativeness of the measuring stations. I think this is a reasonable approach. The only thing I wanted to ask in the previous comment was what the value on the right axis of Figure R1 in the previous response document represents. According to the response document this time, it was added by just some mistake, and I think that it was only necessary to look at the value of the left axis. If this recognition is correct, there is no further request for correction from me.
**Reply:** Thanks for your insightful comments. In the previous response documents, the right axis was added by a mistake. We are very sorry for this mistake.

<About the other replies>
**Comment 2:** Fig.5:Please indicate whether the upper and lower figures are for summer or winter. I think it is also necessary to define the periods for spring, summer, autumn, and winter somewhere in the text.
**Reply:** We agree. In the revised manuscript, we pointed out that the upper and lower panels represent values in summer and winter, respectively. We also defined the periods for spring (March-April-May), summer (June-July-August), autumn (September-October-November), and winter (December-January-February) in the text.

**Comment 3:** Equation (A2) of Appendix A:I think "yi" should be "yij".
**Reply:** We correct it.

**Comment 4:** Fig.3 and Fig.4:I think it is necessary to explain which models the gray line is pointing to.
**Reply:** We add it in the revised manuscript.

We thank the editor for your constructive comments.
Response to the Specific comments.

**Comment 1**:This time the manuscript is much improved according to the previous comments. The reviewer #1 now provides smaller number of minor comments, which should be addressed. Despite previous request to recheck English, the Co-Editor still finds numerous errors and points of revions as follows. They are not all. Thorough checking is mandate, before recommended for publication.
**Reply:** Thanks a lot. We rechecked the English language in the revised manuscript.

(pages and lines are for the cleaned version)

**Comment 1**:1. Abstractm line 1. Rephrase "long-term" as one-year of study is short. For example, Spatio-temporal variations of ozone (O3) and nitrogen oxides (NOx) concentrations from ...
**Reply:** We totally agree and revise it in the new manuscript.

**Comment 2**:Page 2, Line 8. Compared to MICs-Asia II with a study period of four months, the evaluation WITH observations was extended...
**Reply:** We correct it.

**Comment 3**:Page 3, line 26. Page 4, line 13. extra space?
**Reply:** We correct it.

**Comment 4**:Page 4, line 6. The study shows
**Reply:** We correct it.

**Comment 5**:Page 4, lines 10-11. Incomplete sentence (rephrase "have rendering explaining..")
**Reply:** We correct it.

**Comment 6**:Page 4, line 17. have evolved into
**Reply:** We correct it.

**Comment 7**:Page 5, line 10. Use lower case for O3
**Reply:** We correct it.

**Comment 8**:Page 5, line 18. remove extra space and insert space before 2013
**Reply:** We correct it.

**Comment 9**:Page 6, line 4. Remove period character after hPa
**Reply:** We correct it.

**Comment 10**:Page 6, Line 8. remove "are"
**Reply:** We correct it.

**Comment 11**:Page 7, line 4. 2018 not 2017? Check all references.
**Reply:** We correct it.

**Comment 12**::Page 7, lines 24-26 and Page 8, lines 12-14. Similar sentences. Remove one of them
**Reply:** We correct it.
.
**Comment 13**:Page 8, line 29, oxides
**Reply:** We correct it.

**Comment 14**:Page 10, line 19. Insert space after O3
**Reply:** We correct it.

**Comment 15**:Page 10, line 19. Check font of "Most of the"
**Reply:** We correct it.

**Comment 16**:Page 11, lines 6, 17, and 18. Model never measures. Better rephrased to "difficulty in estimating", "in simulating summer patterns", and "simulating winter patterns"
**Reply:** We correct it.

**Comment 17**:Page 12, Line 5. all models showed
**Reply:** We correct it.

**Comment 18**:Page 12, Line 7. High correlations were found
**Reply:** We correct it.

**Comment 19**:Page 12, Line 15, Add space after O3.
**Reply:** We correct it.

**Comment 20**:Page 12, Line 17. Remove extra space after overestimation
**Reply:** We correct it.

**Comment 21**:Page 12, Line 24, Remove "and" before "likely"
**Reply:** We correct it.

**Comment 22**:Page 13, Line 5 and Table 4. Please double check consisntency in the ranges between text and Table. Line 6. For NMBs for EA1, high values from M8-10 are likely omitted. Line 7. Correlation coefficients. Not 0.54 but 0.43, if including M8.
**Reply:** Thanks. We check the statistical numbers in Table 4. We made a mistake for

NO2 NMBs of M8-M10. They should be 14.32%, 32.30% and 10.61%, we omit the character"%". In the revised manuscript, we correct these numbers.

Line 7, we correct the numbers of NMBs in EA3 in the text. "In EA3, correlation coefficients ranged from 0.43-0.72. NMBs and RMSEs except for those of M8 ranged from -0.23-0.46 and 0.90-1.79 ppbv, respectively"

**Comment 23**:Page 13, Line 9. Incomplete sentence.
**Reply:** We correct it.

**Comment 24**:Page 13, Line 11. Not 0.5 but 0.43 when including M2.
**Reply:** We correct it.

**Comment 25**:Page 13, Line 12. Which model has -0.42?
**Reply:** We correct it. "In EA3, correlation coefficients ranged from 0.43-0.72. NMBs and RMSEs except for those of M8 ranged from -0.23-0.46 and 0.91-1.79 ppbv, respectively"

**Comment 26**:Page 13, Line 14. Change font of "the"
**Reply:** We correct it.

**Comment 27**:Page 14, Line 25. The high values
**Reply:** We correct it.

**Comment 28**:Page 15, line 9. Insert space after "lowest levels of"
**Reply:** We correct it.

**Comment 29**:Page 16, Line 3. Lowercase for O3
**Reply:** We correct it.

**Comment 30**:Page 16, lines 10-11. Different sites for II and III for same labels? (e.g., site 4 is Sado-seki in a viewgraph and is Oki for another panel?)
**Reply:** We correct it." site 4: Oki, site 5: Hedo and site 6: Banryu"

**Comment 31**:Page 16, Line 12. remove "been"
**Reply:** We correct it.

**Comment 32**:Page 16, Line 16, enhanced concentrations simulated for
**Reply:** We correct it.

**Comment 33**:Page 17, line 1. replace "be" with "were"
**Reply:** We correct it.

**Comment 34**:Page 17, Line 2 and 5. dry deposition (not depositions)
**Reply:** We correct it.

**Comment 35**:Page 19, Line 15. Change ~ to –
**Reply:** We correct it.

**Comment 36**:Page 19, line 27. remove extra space before Guo
**Reply:** We correct it

**Comment 37**:Page 20, Line 5, add space before 2016
**Reply:** We correct it.

**Comment 38**:Page 20, line 6. Remove "and" before "likely"
**Reply:** We correct it.

**Comment 39**:Page 20, line 1 and 11. Add space before 1996 and before 100-200 m
**Reply:** We correct it.

**Comment 40**:Page 20, line 15. were not was
**Reply:** We correct it.

**Comment 41**:Page 20 Line 19. East Asia
**Reply:** We correct it.

**Comment 42**:Page 28, Remove ACPD for Li, M. et al.
**Reply:** We correct it.

[revised manuscript text omitted]

*Fig.1 Li et al., 2018*

[Figure]

*Fig.2 Li et al., 2018*

[Figure]

*Fig.3 Li et al., 2018*

[Figure]

*Fig.4 Li et al., 2018*

[Figure]

[Figure]

*Fig.5 Li et al., 2018*

[Figure]

*Fig.6 Li et al., 2018*

.

[Figure]

*Fig.7 Li et al., 2018*

[Figure]

*Fig.8 Li et al., 2018*

[Figure]

*Fig.9 Li et al., 2018*

[Figure]

*Fig.10 Li et al., 2018*